



# Indicators of Global Climate Change 2022: Annual update of large-scale indicators of the state of the climate system and the human influence

Piers M. Forster[1], Christopher J. Smith[1,2], Tristram Walsh[3], William F. Lamb[4,1], Matthew D. Palmer[5,6],
Karina von Schuckmann[7], Blair Trewin[8], Myles Allen[3], Robbie Andrew[9], Arlene Birt[10], Alex Borger[11],
Tim Boyer[12], Jiddu A. Broersma[11], Lijing Cheng[13], Frank Dentener[14], Pierre Friedlingstein[15,16], Nathan
Gillett[17], José M. Gutiérrez[18], Johannes Gütschow[19], Mathias Hauser[20], Bradley Hall[21], Masayoshi Ishii[22],
Stuart Jenkins[3], Robin Lamboll[23], Xin Lan[21], June-Yi Lee[24], Colin Morice[5], Christopher Kadow[25], John
Kennedy[26], Rachel Killick[5], Jan Minx[4,1], Vaishali Naik[27], Glen P. Peters[9], Anna Pirani[28], Julia Pongratz[29],
Aurélien Ribes[30], Joeri Rogelj[23], Debbie Rosen[1], Carl-Friedrich Schleussner[31], Sonia I. Seneviratne[20],
Sophie Szopa[32], Peter Thorne[33], Robert Rohde[34], Maisa Rojas Corradi[35], Dominik Schumacher[20], Russell
Vose[36], Kirsten Zickfeld[37], Xuebin Zhang[17], Valerie Masson-Delmotte[38], Panmao Zhai[39]

[1]Priestley Centre, University of Leeds, Leeds, LS2 9JT, UK
[2]International Institute for Applied Systems Analysis (IIASA), Austria
[3]Environmental Change Institute, University of Oxford, UK
[4]Mercator Research Institute on Global Commons and Climate Change (MCC), Berlin, Germany
[5]Met Office Hadley Centre, UK
[6]School of Earth Sciences, University of Bristol, UK
[7]Mercator Ocean international, Toulouse, France
[8]Bureau of Meteorology, Australia
[9]CICERO Center for International Climate Research, Norway
[10]Backgroundstories.com, Minneapolis College of Art and Design
[11]ClimateChangeTracker.org
[12]NOAA National Centers for Environmental Information, Silver Spring, MD, USA
[13]Institute of Atmospheric Physics, Chinese Academy of Sciences, Beijing, China
[14]European Commission, & Joint Research Centre, Institute for Environment and Sustainability, Ispra, Italy
[15]Faculty of Environment, Science and Economy, University of Exeter, UK
[16]Laboratoire de Météorologie Dynamique/Institut Pierre-Simon Laplace, CNRS, Ecole Normale Supérieure/Université PSL, Paris, France
[17]Environment and Climate Change Canada, Canada
[18]Instituto de Física de Cantabria (CSIC-University of Cantabria), Spain
[19]Climate Resource, Australia/Germany
[20]Institute for Atmospheric and Climate Science, Department of Environmental Systems Science, ETH Zurich, Zurich, Switzerland
[21]NOAA Global Monitoring Laboratory, Boulder, CO, USA
[22]Meteorological Research Institute, Tsukuba, Japan
[23]Centre for Environmental Policy, Imperial College London, UK
[24]Research Center for Climate Sciences, Pusan National University and Center for Climate Physics, Institute for Basic Science, Pusan, Republic of Korea
[25]German Climate Computing Center (DKRZ)
[26]No affiliation, independent
[27]NOAA GFDL, Princeton, New Jersey, USA



[28]Université Paris-Saclay, France; CMCC, Italy; Università Cà Foscari, Italy
[29]University of Munich, Germany and Max Planck Institute for Meteorology, Hamburg, Germany
[30]Université de Toulouse, Météo France, CNRS, France
[31]Climate Analytics, Berlin, Germany and Geography Department and IRI THESys, Humboldt-Universität zu Berlin, Berlin, Germany
[32]Laboratoire des Sciences du Climat et de l'Environnement, IPSL, Paris France
[33]ICARUS Climate Research Centre, Maynooth University, Maynooth, Ireland
[34]Berkeley Earth, Berkeley, CA, USA
[35]University of Chile, Santiago, Chile
[36]NOAA's National Centers for Environmental Information (NCEI), Asheville, NC, USA
[37]Simon Fraser University, Vancouver, Canada
[38]Laboratoire des Sciences du Climat et de l'Environnement (LSCE) / Institut Pierre Simon Laplace (IPSL), CEA-CNRS-
UVSQ (UMR8212), Université Paris-Saclay, Gif-sur-Yvette, France
[39]Chinese Academy of Meteorological Sciences, Beijing, China

*Correspondence to*: Piers. M. Forster (p.m.forster@leeds.ac.uk)

**Abstract.** Intergovernmental Panel on Climate Change (IPCC) assessments are the trusted source of scientific evidence for climate negotiations taking place under the United Nations Framework Convention on Climate Change (UNFCCC), including
the first global stocktake under the Paris Agreement that will conclude at COP28 in December 2023. Evidence-based decision making needs to be informed by up-to-date and timely information on key indicators of the state of the climate system and of the human influence on the global climate system. However, successive IPCC reports are published at intervals of 5-10 years, creating potential for an information gap between report cycles.

We base this update on the assessment methods used in the IPCC Sixth Assessment Report (AR6) Working Group One (WGI) report, updating the monitoring datasets and to produce updated estimates for key climate indicators including emissions, greenhouse gas concentrations, radiative forcing, surface temperature changes, the Earth's energy imbalance, warming attributed to human activities, the remaining carbon budget and estimates of global temperature extremes. The purpose of this effort, grounded in an open data, open science approach, is to make annually updated reliable global climate indicators
available in the public domain (https://doi.org/10.5281/zenodo.7883758, Smith et al., 2023). As they are traceable and consistent with IPCC report methods, they can be trusted by all parties involved in UNFCCC negotiations and help convey wider understanding of the latest knowledge of the climate system and its direction of travel.

The indicators show that human induced warming reached 1.14 [0.9 to 1.4] °C over the 2013-2022 period and 1.26 [1.0 to 1.6]
°C in 2022. Human induced warming is increasing at an unprecedented rate of over 0.2 °C per decade. This high rate of warming is caused by a combination of greenhouse gas emissions being at an all-time high of $57 \pm 5.6$ GtCO₂e over the last decade, as well as reductions in the strength of aerosol cooling. Despite this, there are signs that emission levels are starting to



stabilise, and we can hope that a continued series of these annual updates might track a real-world change of direction for the climate over this critical decade.

## 1 Introduction

Increased greenhouse gas concentrations combined with reductions in aerosol pollution have led to rapid increases in human induced effective radiative forcing and atmosphere, land, cryosphere and ocean warming (Gulev et al., 2021). This in turn has led to an intensification of many weather and climate extremes, particularly more frequent and more intense hot extremes, and heavy precipitation across most regions of the world (Seneviratne et al., 2021). Given the speed of recent change, and the need for evidence-based decision-making, this Indicators of Global Climate Change (IGCC) update is proposed to assemble the latest scientific understanding on the current state of the climate system, how it is evolving and the human influence to support policymakers whilst the next IPCC assessment is under preparation.

This update analyses and integrates international efforts, especially those of the Global Climate Observing System (GCOS) Essential Climate Variables programme and the World Meteorological Organization's (WMO) Global Atmospheric Watch programme. Annual state of the climate reports are released by WMO which use much of the same data analysed here for surface temperature and energy budget trends. The Bulletin of American Meteorological Society release annual State of the Climate reports covering many essential variables including temperature. However, these reports focus on statistics from the previous year and make slightly different choices over datasets and analysis compared to IPCC (see Sect. 5). The Global Carbon Project publishes updated carbon dioxide and methane emission datasets which are used directly in this report. There is no current report that updates all the necessary datasets to make an annual assessment of the human influence on surface temperature - and this is the goal here.

The update is based on methodologies of key climate indicators assessed by the IPCC Sixth Assessment Report (AR6) of the physical science basis of climate change (WGI report IPCC (2021a) as well as Chapter 2 of the WGIII report (Dhakal et al., 2022), and is aligned with the efforts initiated in AR6 to implement FAIR principles for reproducibility and reusability (Pirani et al. 2022, Iturbide et al. 2022). We trace methodologies to these IPCC reports, but as calibrations are revised and science moves forward, we update methods where necessary. However, we do this as transparently as possible to distinguish methodological differences from physical evolution. IPCC reports make a much wider assessment of the science and methodologies - we do not attempt to reproduce the comprehensive nature of these IPCC assessments here.



The COP21 Paris Agreement of 2015 expressly sets out to limit global warming levels through greenhouse gas emission reduction commitments in Articles 2 and 4 respectively. Article 2.1.a sets the goal of holding global temperature increase to well below 2°C above pre-industrial levels, and pursuing efforts to limit the increase to 1.5°C; Article 4.1 states the aim for global greenhouse gas emissions (GHGs) to peak as soon as possible, and to reach a balance between anthropogenic emissions by sources and removals by sinks of GHGs in the second half of the century. Article 2 also sets out clear targets for adaptation (Article 2.1b) and implementation (Article 2.1c). Establishing policies to effectively support efforts to meet these aims and commitments requires reliable indicators of both the state of the climate system and the human influence on climate.

Both COP26 (Glasgow) and COP27 (Sharm El-Sheikh) also "recognized[s] the importance of the best available science for effective climate action" (UNFCCC, 2022a,b). COP27 in 2022 reiterated its invitation to Parties to consider further actions to reduce by 2030 non-carbon dioxide greenhouse gas emissions, including methane. A global stocktake to be held every five years, starting in 2023, has been established under the Paris Agreement to evaluate the collective progress of countries' actions in the implementation of the Paris Agreement and its long-term goals. The IPCC assessment of the physical science basis provides a wide range of information with relevance for the global stocktake, complementing other products from AR6. The now complete AR6 cycle updated GHG emissions and concentrations, the current state of the climate, near-term and long-term projections of global warming and of the climate system, the attribution of extreme events, and remaining carbon budgets.

The 2015 COP21 Decision invited the IPCC to prepare a special report on the impacts of 1.5°C and related greenhouse gas emission pathways to help inform its work (UNFCCC, 2015). The resulting IPCC Special Report on Global Warming of 1.5°C (SR1.5), published in 2018, provided an assessment of the level of human-induced warming and cumulative emissions to date (Allen et al., 2018) and the remaining carbon budget (Rogelj et al., 2018) to support the evidence base on how the world is progressing in terms of meeting the Paris Agreement's Article 2. The AR6 WGI Report, published in 2021, assessed past, current and future changes of these and other key global climate indicators, as well as undertaking an assessment of the Earth's energy budget. Given the current rates of change and the likelihood of reaching 1.5°C of global warming in the first half of the 2030s (Lee et al., 2021b; Lee et al., 2023; Riahi et al., 2022), it is important to have robust, trusted, and also timely climate indicators in the public domain to form an evidence base for effective science-based decision making. Cross-Chapter Box 1.1, Table 1 in Chapter 1 of the AR6 report (Chen et al., 2021) maps how the material assessed by WGI may be relevant for the global stocktake.

When making their assessments, authors of IPCC reports assess published literature, but also apply established published analysis methods to assessed datasets, such as that produced by the latest climate model intercomparison projects (Lee et al.,



2021b). The authors combine and analyze both model and observational data as part of their expert assessment, making assessments of the trustworthiness and error characteristics of different datasets. It is this synthetic analysis by IPCC authors that derives the estimates of key climate indicators. It is these same assessed approaches that we are implementing here to provide updates. The same approach, using the same datasets (updated by 2 years) and methods as employed in WGI, was used in the AR6 SYR (2023) report to provide an updated assessment of the latest atmospheric well mixed greenhouse gas concentrations (up to 2021) and decadal average change in global surface temperature (+1.15°C [1.00°C–1.25°C] in 2013-2022 for global surface temperature). However, the assessment of human-induced warming was not updated (and therefore only covers warming up to the decade 2010-2019), nor was the remaining carbon budget updated, so the related information in the 2023 SYR report remained based on data up to the end of 2019.

In this work, we focus on providing updated estimates of key global indicators of the state of climate: the Earth's heat inventory, human-induced warming and the remaining carbon budget. To do this requires updates to emissions, greenhouse gas concentrations, effective radiative forcing, energy imbalance and surface temperature change, which are all important indicators of the direction of travel, giving important insights to the magnitude and the pace of global warming. This paper provides the basis for a dashboard of climate indicators grounded in IPCC methodologies and directly traceable to reports published as part of the AR6 cycle. We employ datasets that can be updated on a regular basis between the publication of IPCC reports. Note that there are other similar initiatives underway to update other AR6 cycle products; for example, the evolution of the WGI Interactive Atlas (Gutierrez et al. 2021) is being developed under the Copernicus Climate Change Service (C3S) and has potential connections and synergies with this initiative that will be explored in the future.

We track how specific global climate indicators have changed since reported in AR6. It breaks the change down into components from the march of time and any revisions to methods and/or data. For this first report, the methodologies are either identical or very close to those employed in AR6, keeping our results as consistent as possible. There are places where we need to depart as indicated in the text. These occasions are when a particular dataset is not updateable through 2022 and an alternative approach has had to be used.

Our longer-term ambition is to rigorously track both real system change and methodological improvements between IPCC report cycles, thereby building consistency and awareness. An example of why tracking methodological change is important, was the shift in the historical baseline used for global surface temperatures between SR1.5 and AR6. Datasets and methods of evaluating global temperature changes altered between the fifth (AR5) assessment report and AR6, leading to a small shift in the historical temperature baseline. This was reflected in changes between AR5 and AR6, whereas SR1.5 chose to broadly



follow AR5 methods (see AR6 WGI Cross Chapter Box 2.1, Gulev et al., 2021). Annual updates provide forewarning of
possible future methodological shifts that subsequent IPCC reports may make as science advances and can detail their impact
on perceived trends.

We adopt the Global Carbon Budget ethos of a community-wide inclusive effort that synthesises work from  across a large
and diverse global scientific community in a timely fashion (Friedlingstein et al., 2022). Like the Global Carbon Budget, this
initiative arises from the international science community to establish a knowledge base to support policy debate and action to
slow down and ultimately stop the increase of greenhouse gases in the atmosphere. The update is focussed on building from
emissions towards estimates of human-induced warming and the remaining carbon budget. Emissions (Sect. 2) and GHG
concentrations (Sect.  3) are used to develop updated estimates of effective radiative forcing (Sect.  4). Observations of global
surface temperature change (Sect.  5) and Earth's energy imbalance (Sect.  6) are key global indicators of a warming world.
The global temperature change is formally attributed to human activity in Sect. 7, which tracks human-induced warming.
Section 8 updates the remaining carbon budget to policy-relevant temperature thresholds. Section 9 estimates preliminary
global-scale indicators of climate extremes. These changes in extremes are not directly related to the other indicators but are
included to showcase the possibility of extending the indicator set in future years.

An important purpose of the exercise is to make these indicators widely available and understood. Plans for a web dashboard
are discussed in Sect. 10, code and data availability in Sect. 11 and conclusions presented in Sect. 12. Data is available at
https://doi.org/10.5281/zenodo.7883758, Smith et al. (2023).

## 2. Emissions

Historic emissions from human activity were assessed in both AR6 WGI and WGIII. Chapter 5 of WGI assessed $CO_2$ and $CH_4$
emissions in the context of the carbon cycle (Canadell et. al., 2021). Chapter 6 of WGI assessed emissions in the context of
understanding the climate and air quality impact of short-lived climate forcers (Szopa et al., 2021). Chapter 2 of WGIII
published one year later (Dhakal et al., 2022) looked at the sectoral sources of emissions and gave the most up to date
understanding of the current level of emissions. This section based its methods and data on those employed in this WGIII
chapter.



## 2.1 Methods of estimating greenhouse gas emissions changes

Like in AR6 WGIII, net GHG emissions in this paper refer to releases of GHG from anthropogenic sources minus removals by anthropogenic sinks, for those species of gases that are reported under the common reporting format of the UNFCCC. This includes $CO_2$ emissions from fossil fuels and industry ($CO_2$-FFI), net $CO_2$ emissions from land use, land use change and forestry ($CO_2$-LULUCF), $CH_4$, $N_2O$ and fluorinated gas (F-gas) emissions. $CO_2$-FFI mainly comprises fossil-fuel combustion emissions, as well as emissions from industrial processes such as cement production. $CO_2$-LULUCF - also known as land use change emissions - is mainly driven by deforestation, but also includes anthropogenic removals on land from afforestation and reforestation, as well as emissions and removals from other land-use change and land management activities such as peat drainage or forestry. The non-$CO_2$ GHG emissions - $CH_4$, $N_2O$ and F-gases emissions - are linked to the fossil-fuel extraction, agriculture and industry sectors.

Global regulatory conventions have led to a two-fold categorisation of F-gas emissions (also known as halogenated gases). Under UNFCCC accounting, countries record emissions of hydrofluorocarbons (HFCs), perfluorocarbons (PFCs), sulphur hexafluoride (SF6), and nitrogen trifluoride (NF3) - hereinafter "UNFCCC F-gases". However, national inventories tend to exclude halons, chlorofluorocarbons (CFCs) and hydrochlorofluorocarbons (HCFCs) - hereinafter "ODS (Ozone Depleting Substances) F-gases" - as they have been initially regulated under the Montreal protocol and its amendments. In line with the WGIII assessment, ODS-F-gases and other substances, including ozone and aerosols, are not included in our GHG emissions reporting, but are included in subsequent assessment of concentration, effective radiative forcing, human-induced warming, carbon budgets and climate impacts in line with the WGI assessment.

Each category of GHG emissions included here is covered by varying primary sources and datasets, with significant differences in uncertainties. Although many datasets cover individual categories, few extend across multiple categories, and only a minority have frequent and timely update schedules. Notable datasets include the Global Carbon Budget (GCB; Friedlingstein et al., 2022), which covers $CO_2$-FFI and $CO_2$-LULUCF; the Emissions Database for Global Atmospheric Research (EDGAR; Crippa et al., 2022) and the Potsdam Real-time Integrated Model for probabilistic Assessment of emissions Paths (PRIMAP-hist; Gütschow et al., 2016; Gütschow and Pflüger 2023), which cover $CO_2$-FFI, $CH_4$, $N_2O$ and UNFCCC F-gases; and the Community Emissions Data system (CEDS; O'Rourke et al., 2021), which covers $CO_2$-FFI, $CH_4$, and $N_2O$.

In AR6 WGIII, total net GHG emissions were calculated as the sum of $CO_2$-FFI, $CH_4$, $N_2O$ and UNFCCC F-gases from EDGAR, and net $CO_2$-LULUCF emissions from the GCB. Net $CO_2$-LULUCF emissions followed the GCB convention and were derived from the average of 3 bookkeeping models (Hansis et al., 2015; Houghton and Nassikas 2017; Gasser et al.,



2020). Version 6 of EDGAR was used (with a fast-track methodology applied for the final year of data - 2019), alongside the 2020 version of the GCB (Friedlingstein et al., 2020). $CH_4$ and $N_2O$ emissions from biomass combustion from the Global Fire Emissions Database (GFED; Van Der Werf 2017) were added to EDGAR. $CO_2$-equivalent emissions were calculated using global warming potentials with a 100-year time horizon AR6 WGI Chapter 7 (Forster et al., 2021). Uncertainty ranges were

based on a comparative assessment of available data and expert judgement, corresponding to a 90% confidence interval (Minx et al., 2021): ±8% for $CO_2$-FFI, ±70% for $CO_2$-LULUCF, ±30% for $CH_4$ and F-gases, and ±60% for $N_2O$. (Note that the GCB assesses uncertainties for $CO_2$-FFI as ±5%, and uses an absolute uncertainty range of ±2.6 $GtCO_2$ for $CO_2$-LULUCF, both corresponding to a 95% confidence interval; Friedlingstein et al., 2022). The aggregate uncertainty for GHG emissions was computed as the square root of the sum of squared uncertainties for each gas. Reflecting these uncertainties, AR6 WGIII

reported emissions to two significant figures only. Uncertainties in GWP100 metrics were not applied (Minx et al., 2021).

This manuscript tracks the same compilation of GHGs as in AR6 WGIII. We follow the same approach for estimating uncertainties and $CO_2$-equivalent emissions. We also use the same type of data sources, but make important changes to their specific selection of data sources to further improve the quality of the data as suggested in the knowledge gap discussion of

the WGIII report (Dhakal et al., 2022). Instead of using EDGAR data, we use GCB data for $CO_2$-FFI, PRIMAP-hist data for $CH_4$ and $N_2O$, and atmospheric concentrations with best-estimate lifetimes for UNFCCC F-gas emissions (Hodnebrog et al., 2020b). As in AR6 WGIII we use GFED for biomass combustion $CH_4$ and $N_2O$ emissions, and GCB for net $CO_2$-LULUCF emissions, taking the average of 3 bookkeeping models.

There are three reasons for these specific data choices. First, national greenhouse gas emissions inventories tend to use improved, higher-tier methods for estimating emissions fluxes than global inventories such as EDGAR or CEDS (Dhakal et al., 2022; Minx et al., 2021). As GCB and PRIMAP-hist integrate the most recent national inventory submissions to the UNFCCC, selecting these databases makes best use of the investments countries have made into data gathering infrastructures. Second, comprehensive reporting of F-gas emissions has remained challenging in national inventories (see Minx et al., 2021;

Dhakal et al., 2022). However, as global F-gas concentrations are entirely anthropogenic substances they can be measured effectively and reliably in the atmosphere. We therefore follow the AR6 WGI approach in making use of direct atmospheric observations. Third, the choice of GCB data for $CO_2$-FFI means we can integrate its projection of that year's $CO_2$ emissions at the time of publication (i.e., for 2022). No other dataset except GCB provides projections of $CO_2$ emissions on this timeframe. At this point in the publication cycle (mid-year), the other chosen sources provide data points with a two-year time lag (i.e.,

for 2021). While these data choices inform our overall assessment of GHG emissions, we provide a comparison across datasets





for each emissions category, as well as between our estimates and an estimate derived from AR6 WGIII-like databases (i.e., EDGAR for $CO_2$-FFI and non-$CO_2$ GHG emissions, GCB for $CO_2$-LULUCF).

## 2.2 Updated global greenhouse gas emissions

Total global GHG emissions reached $57 \pm 5.6$ $GtCO_2e$ in 2021. The main contributing sources were $CO_2$-FFI ($37 \pm 3$ $GtCO_2$),

$CO_2$-LULUCF ($3.9 \pm 2.8$ $GtCO_2$), $CH_4$ ($11 \pm 3.2$ $GtCO_2e$), $N_2O$ ($3.3 \pm 2$ $GtCO_2e$) and F-gas emissions ($2 \pm 0.59$ $GtCO_2e$). GHG emissions rebounded in 2021, following a significant single year decline during the COVID-19 induced lockdowns of 2020. Prior to this event in 2019, emissions were $57 \pm 5.7$ $GtCO_2e$ - i.e. almost the same level as in 2021. Initial projections indicate that $CO_2$ emissions from fossil fuel and industry and land use change remained stable in 2022, at $37 \pm 3$ $GtCO_2$ and $3.9 \pm 2.8$ $GtCO_2$, respectively (Friedlingstein et al., 2022). Note that ODS-F-gases such as chlorofluorocarbons and

hydrochlorofluorocarbons are excluded from national GHG emissions inventories. For consistency with AR6, they are also excluded here. Including them here would increase total global GHG emissions by 1.6Gt $GtCO_2e$ in 2021.

Average GHG emissions for the decade 2012-2021 were $56 \pm 5.6$ $GtCO_2e$. Average decadal GHG emissions have increased steadily since the 1970s across all major groups of GHG, driven primarily by increasing $CO_2$ emissions from fossil fuel and

industry, but also rising emissions of $CH_4$ and $N_2O$. UNFCCC F-gas emissions have grown more rapidly than other greenhouse gases reported under the UNFCCC, but from low levels. By contrast, ODS F-gas emissions have declined substantially since the 1990s. Both the magnitude and trend of $CO_2$ emissions from land use change remain highly uncertain, with the latest data indicating a relatively stable net flux between 4-5 $GtCO_2$ /yr for the past few decades.

AR6 WGIII reported total net anthropogenic emissions of $59 \pm 6.6$ $GtCO_2e$ in 2019, and decadal average emissions of $56 \pm 6.0$ $GtCO_2e$ from 2010-2019. By comparison, our estimates here for the AR6 period sum to $57 \pm 5.7$ $GtCO_2e$ in 2019, and $55 \pm 5.6$ $GtCO_2e$ for the same decade (2010-2019). The difference between these figures is primarily driven by the substantial revision in GCB $CO_2$-LULUCF estimates: between the 2020 version (used in AR6 WGIII) of 6.6 $GtCO_2$ and the 2022 version (used here) of 4.6 $GtCO_2$. The main reason for this downward revision comes from updated estimates of agricultural areas by the

FAO and uses multi-annual land-cover maps from satellite remote sensing, leading to lower emissions from cropland expansion, particularly in the tropical regions. It is important to note that this change is not a reflection of an improved estimation methodology, but an adjustment of the methodology due to changes in the available input data. Second, there are smaller differences in the estimates from the described use of better datasets in this study. However, as shown in Figure 1 below these are relatively small in magnitude and the direction of these differences depend on the gases considered (increases:





N₂O (+0.5 GtCO₂e), UNFCCC-F-gases (+ 0.48 GtCO₂e); and decreases: CH₄ (-0.001 GtCO₂e), CO2-FFI (-0.8 GtCO₂e)). Overall, excluding the change due to CO₂-LULUCF, this only impacts the total GHG emissions estimate by 0.19 GtCO₂e.

**Figure 1: Annual global anthropogenic greenhouse gas emissions by source, 1970-2021.** Refer to Sect. 2.1 for a list of datasets. Starred
datasets (*) indicate the sources used to compile global total greenhouse gas emissions in panel a. CO₂ equivalent emissions in panels a and f are calculated using GWPs with a 100-year time horizon from the AR6 WGI Chapter 7 (Forster et al., 2021). F-gas emissions





**in panel a comprise only UNFCCC F-gas emissions (see Sect. 2.1 for a list of species). Not shown in panels d and e are biomass combustion emissions from GFED (Van Der Werf 2017), which are included in the aggregate estimate in panel a.**


**Table 1: Global anthropogenic greenhouse gas emissions by source and decade.**

| Gt CO$_2$e | 1970-1979 | 1980-1989 | 1990-1999 | 2000-2009 | 2012-2021 | 2021 | 2022 (projection) |
|---|---|---|---|---|---|---|---|
| **GHG** | 31±4.2 | 36±4.6 | 41±5.1 | 47±5.3 | 56±5.6 | 57±5.6 | |
| **CO$_2$-FFI** | 17±1.4 | 20±1.6 | 24±1.9 | 29±2.3 | 36±2.9 | 37±3 | 37±3 |
| **CO$_2$-LUCF** | 4.4±3.1 | 4.8±3.4 | 5.3±3.7 | 5±3.5 | 4.5±3.2 | 3.9±2.8 | 3.9±2.8 |
| **CH$_4$** | 6.9±2.1 | 7.6±2.3 | 8.4±2.5 | 9±2.7 | 10±3.1 | 11±3.2 | |
| **N$_2$O** | 2.1±1.3 | 2.4±1.4 | 2.6±1.5 | 2.8±1.7 | 3.1±1.9 | 3.3±2 | |
| **UNFCCC F-gases** | 0.58±0.17 | 0.78±0.23 | 0.77±0.23 | 1±0.3 | 1.7±0.5 | 2±0.59 | |

**Notes: All numbers refer to decadal averages, except for annual estimates in 2021 and 2022. CO$_2$ equivalent emissions are calculated using GWP with a 100-year time horizon from AR6 WGI Chapter 7 (Forster et al., 2021). Projections of non-CO$_2$ GHG emissions in 2022 remain unavailable at the time of publication. Uncertainties are ±8% for CO$_2$-FFI, ±70% for CO$_2$-LULUCF, ±30% for CH$_4$ and F-gases, and ±60% for N$_2$O, corresponding to a 90% confidence interval. ODS F-gases are excluded, as noted in Sect. 2.1.**

**2.3 Non-methane short lived climate forcers**

In addition to GHG emissions, we provide an update of anthropogenic emissions of non-methane short-lived climate forcers (SLCFs) (SO$_2$, BC, OC, NO$_x$, VOCs, CO and NH$_3$). Updating emissions of many short-lived climate forcing agents to 2022 based on established datasets is not possible as compiling global data can take several years. Yet, as SLCF emissions are needed in this paper to update effective radiative forcing (ERF) estimates through 2022, updated emission datasets, where they are available, are combined with projected data to make SLCF emission time series complete.


As in Dhakal et al. (2022), sectoral emissions of SLCFs are derived from two sources. For fossil fuel, industrial, waste and agricultural sectors, we use the CEDS dataset that provided SLCF emissions for CMIP6 (Hoesly et al., 2018). CEDS provides



global emissions totals from 1750 to 2019 in its most recent version (O'Rourke et al., 2021). No CEDS emissions data is yet available beyond 2019. As a first estimate, the SLCF emissions time series are extrapolated to 2022 using the "two-year blip"

scenario (Forster et al., 2020) of global emissions suppressed by the economic slowdown due to COVID-19. These projections are proxy estimates from Google and Apple mobility data over 2020, and assume a slow return to pre-pandemic emissions activity levels by 2022. Other near-real time emissions estimates covering the COVID-19 pandemic era tend to show less of an emissions reduction than the two-year blip scenario (Guevara et al., 2023). It should be stressed that accurate quantification of emission during this period is not possible.


For biomass burning SLCF emissions we follow AR6 WGIII (Dhakal et al., 2022) and use GFED for 1997 to 2022, with the dataset extended back to 1750 for CMIP6 (van Marle et al., 2017). Estimates from 2017 to 2022 are provisional. The potential for both sources of emissions data to be updated in future versions exist. Other natural emissions, which are important for gauging some SLCF concentrations, are considered as constant in the context of calculating concentrations and ERF.


Estimated emissions used here are based on a combination of GFED emissions for biomass-burning emissions and CEDS up until 2019 extended with the "two-year blip" scenario for fossil, agricultural, industrial and waste sectors. Under this scenario, emissions of all SLCFs are reduced in 2022 relative to 2019 (Table 2). As described in Sect. 4, this has implications for several categories of anthropogenic radiative forcing. Trends in SLCFs emissions are spatially heterogeneous (Szopa et al., 2021) with

strong shifts in the geographical distribution of emissions over the 2010-2019 decade. Very different lockdown measures have been applied for COVID around the world resulting in various length and intensity of activity reductions and effect on air pollutant emissions (Sokhi et al., 2021). SLCF emissions have been seen to return to their pre-COVID levels by 2022 in some regions, sometimes with rebound effect, but not in all (Putaud et al., 2023, Lonsdale and Sun, 2023) but quantification at the global scale is not yet available.


Uncertainties in the emissions underlying data are difficult to quantify. From the non-biomass burning sectors they are estimated to be smallest for $SO_2$ (±14%), largest for BC (a factor of two), and intermediate for other species (Smith et al., 2011; Bond et al., 2013; Hoesly et al., 2018). Uncertainties are also likely to increase backwards in time (Hoesly et al., 2018), and in the most recent years. The estimates of non-biomass burning emissions for 2020, 2021 and 2022 are highly uncertain owing

to the use of proxy activity data and scenario extension. We do not provide a formal assessment of emissions uncertainty here as uncertainties in underlying datasets are not routinely quantified. Future updates of CEDS are expected to include uncertainties (Hoesly et al., 2018). Even though trends over recent years are uncertain, the general decline in SLCF emissions derived is supported by aerosol optical depth measurements (e.g. Quaas et al., 2022).





**Table 2: Emissions of the major SLCFs in 1750, 2019 and 2022**

| Compound Species | 1750 emissions (Tg yr⁻¹) | 2019 emissions (Tg yr⁻¹) | 2022 emissions (Tg yr⁻¹) |
|---|---|---|---|
| Sulphur dioxide ($SO_2$) + sulphate ($SO_4^{2-}$) | 0.3 | 85.9 | 76.9 |
| Black carbon (BC) | 2.1 | 7.8 | 6.7 |
| Organic carbon (OC) | 15.4 | 34.7 | 26.0 |
| Ammonia ($NH_3$) | 6.6 | 66.5 | 65.3 |
| Oxides of nitrogen (NOx) | 19.4 | 142.9 | 131.8 |
| Volatile organic compounds (VOCs) | 60.6 | 227.2 | 189.6 |
| Carbon monoxide (CO) | 348.4 | 937.8 | 764.1 |

*Notes. Emissions of $SO_2$ + $SO_4^{2-}$ use $SO_2$ molecular weights. Emissions of NOx use $NO_2$ molecular weights. VOCs are for the total mass.*

## 3 Well-mixed greenhouse gas concentrations

AR6 Working Group I assessed well mixed GHG concentrations in Chapter 2 (Gulev et al., 2021) and additionally provided a dataset of concentrations of 52 well-mixed GHGs from 1750 to 2019 in its Annex III (IPCC, 2021c). Footnotes in AR6 SYR

updated $CO_2$, $CH_4$ and $N_2O$ concentrations to 2021 (Lee et al., 2023). In this update we extended the record to 2022 for all 52 gases.

Ozone is an important greenhouse gas with strong regional variation both in the stratosphere and troposphere (Szopa et al., 2021). Its ERF arising from its regional distribution is assessed in Sect. 4, but following AR6 convention is not included with

the GHGs discussed here. Other non-methane SLCFs are heterogeneously distributed in the atmosphere and are also not typically reported in terms of a globally averaged concentration. Globally averaged concentrations for these are normally model derived, supplemented by local monitoring networks and satellite data (Szopa et al., 2021).

As in AR6, $CO_2$ concentrations are taken from the NOAA Global Monitoring Laboratory (GML) and updated through 2022

(Lan et al., 2023a). Although, here CO2 is reported on the updated WMO-CO2-X2019 scale, whereas in AR6, values were reported on the WMO-CO2-X2007 scale. This improved calibration increases $CO_2$ concentrations by around 0.2 ppm (Hall et al., 2021). In AR6, $CH_4$ and $N_2O$ were reported as the average from NOAA and AGAGE global networks. For 2022 as updated AGAGE data is not currently available, we used only NOAA data [Lan et al., 2023b], and multiplied $N_2O$ by 1.0007 to be consistent with a NOAA-AGAGE average. NOAA $CH_4$ in 2022 was used without adjustment since the NOAA and AGAGE

global means $CH_4$ are consistent within 2 ppb. Mixing ratio uncertainties for 2022 are assumed to be similar to 2019, and we adopt the same uncertainties as assessed in AR6 WGI.

Many halogenated greenhouse gases are reported on a global mean basis from NOAA and/or AGAGE until 2020 or 2021 ($SF_6$ is available in the NOAA dataset up to 2022). Where both NOAA and AGAGE data are used for the same gas, we take a mean

of the two datasets. Where both networks are used and the last full year of data availability is different, the difference between the dataset mean and the dataset with the longer time series in this last year is used as an additive offset to the dataset with the longer time series. Some obvious inconsistencies are removed such as sudden changes in concentrations when missing data is reported as zero.

Some of the more minor halogenated gases are not part of the NOAA or AGAGE operational network and are currently only reported in literature sources until 2019, or possibly 2015 (Droste et al., 2020; Laube et al., 2014; Schoenenberger et al., 2015; Simmonds et al., 2017; Vollmer et al., 2018). Concentrations of gases where 2022 data is not yet available are extrapolated forwards to 2022 using the average growth rate over the last 5 years of available data. These assumptions have an imperceptible effect on the total ERF assessed in Sect. 4.






The global surface mean mixing ratios of $CO_2$, $CH_4$ and $N_2O$ in 2022 were 417.1 [± *0.4*] ppm, 1911.9 [± *3.3*] ppb and 335.9 [± *0.4*] ppb (table 3). Concentrations of all three major GHGs have increased from 2019 reported in AR6 WGI, which were 410.1 [± 0.36] ppm for $CO_2$, 1866.3 [± 3.2] ppb for $CH_4$ and 332.1 [± 0.7] ppb for $N_2O$. Note, AR6 SYR quoted updated values but to less precision. These are not given here to avoid the perception of an inconsistency. Concentrations of most categories of

halogenated GHGs have increased from 2019 to 2022, from 109.4 to 114.2 ppt $CF_4$-eq for PFCs, 237.1 ppt to 287.2 ppt HFC-134a-eq for HFCs, 9.9 ppt to 11.0 ppt for $SF_6$ and 2.1 to 2.8 ppt for $NF_3$. Only Montreal Protocol halogenated GHGs have decreased in concentration, from 1031.9 ppt in 2019 to 1016.6 ppt in 2022, demonstrating the continued success of the Montreal Protocol. Although even here, concentrations of some minor CFCs are rising (see also Western et al. 2023). In this update we employ AR6 derived uncertainty estimates and do not perform a new assessment.


**Table 3: Annual mean concentrations of well-mixed greenhouse gases in 2022, 2019, 1850 and 1750. Except for $CO_2$, $CH_4$ and $N_2O$, concentrations all are in parts per trillion by volume [ppt].**

| Greenhouse gas | 2022 | 2019 | 1850 | 1750 |
|---|---|---|---|---|
| $CO_2$ [ppm] | 417.1 | 410.1 | 285.5 | 278.3 |
| $CH_4$ [ppb] | 1911.9 | 1866.3 | 807.6 | 729.2 |
| $N_2O$ [ppb] | 335.9 | 332.1 | 272.1 | 270.1 |
| $NF_3$ | 2.7 | 2.1 | 0.0 | 0.0 |
| $SF_6$ | 11.0 | 9.9 | 0.0 | 0.0 |
| $SO_2F_2$ | 2.8 | 2.5 | 0.0 | 0.0 |
| HFCs as HFC-134a-eq | 287.2 | 237.7 | 0.0 | 0.0 |
| HFC-23 | 36.1 | 32.5 | 0.0 | 0.0 |
| HFC-32 | 31.1 | 20.4 | 0.0 | 0.0 |
| HFC-125 | 39.7 | 29.5 | 0.0 | 0.0 |
| HFC-134a | 124.5 | 107.6 | 0.0 | 0.0 |



| | | | | |
|---|---|---|---|---|
| **HFC-143a** | 28.9 | 24.0 | 0.0 | 0.0 |
| **HFC-152a** | 7.5 | 7.2 | 0.0 | 0.0 |
| **HFC-227ea** | 2.1 | 1.6 | 0.0 | 0.0 |
| **HFC-236fa** | 0.2 | 0.2 | 0.0 | 0.0 |
| **HFC-245fa** | 3.7 | 3.1 | 0.0 | 0.0 |
| **HFC-365mfc** | 1.2 | 1.1 | 0.0 | 0.0 |
| **HFC-43-10mee** | 0.3 | 0.3 | 0.0 | 0.0 |
| **PFCs as $CF_4$-eq** | 114.2 | 109.4 | 34.0 | 34.0 |
| **$CF_4$** | 88.4 | 85.6 | 34.0 | 34.0 |
| **$C_2F_6$** | 5.1 | 4.8 | 0.0 | 0.0 |
| **$C_3F_8$** | 0.7 | 0.7 | 0.0 | 0.0 |
| **$c\text{-}C_4F_8$** | 1.9 | 1.8 | 0.0 | 0.0 |
| **$n\text{-}C_4F_{10}$** | 0.2 | 0.2 | 0.0 | 0.0 |
| **$n\text{-}C_5F_{12}$** | 0.2 | 0.1 | 0.0 | 0.0 |
| **$n\text{-}C_6F_{14}$** | 0.2 | 0.2 | 0.0 | 0.0 |
| **$i\text{-}C_6F_{14}$** | 0.1 | 0.1 | 0.0 | 0.0 |
| **$C_7F_{16}$** | 0.1 | 0.1 | 0.0 | 0.0 |
| **$C_8F_{18}$** | 0.1 | 0.1 | 0.0 | 0.0 |
| **Montreal gases as CFC-12-eq** | 1016.6 | 1031.8 | 8.5 | 8.5 |
| **CFC-11** | 219.6 | 226.2 | 0.0 | 0.0 |



| | | | | |
|---|---|---|---|---|
| **CFC-12** | 493.3 | 502.9 | 0.0 | 0.0 |
| **CFC-112** | 0.4 | 0.4 | 0.0 | 0.0 |
| **CFC-112a** | 0.1 | 0.1 | 0.0 | 0.0 |
| **CFC-13** | 3.4 | 3.3 | 0.0 | 0.0 |
| **CFC-113** | 68.2 | 69.8 | 0.0 | 0.0 |
| **CFC-113a** | 1.0 | 0.9 | 0.0 | 0.0 |
| **CFC-114** | 16.3 | 16.3 | 0.0 | 0.0 |
| **CFC-114a** | 1.0 | 1.0 | 0.0 | 0.0 |
| **CFC-115** | 8.8 | 8.7 | 0.0 | 0.0 |
| **HCFC-22** | 251.8 | 246.8 | 0.0 | 0.0 |
| **HCFC-31** | 0.1 | 0.1 | 0.0 | 0.0 |
| **HCFC-124** | 0.9 | 1.0 | 0.0 | 0.0 |
| **HCFC-133a** | 0.5 | 0.4 | 0.0 | 0.0 |
| **HCFC-141b** | 24.6 | 24.4 | 0.0 | 0.0 |
| **HCFC-142b** | 21.9 | 22.2 | 0.0 | 0.0 |
| **$CH_3CCl_3$** | 0.9 | 1.6 | 0.0 | 0.0 |
| **$CCl_4$** | 74.0 | 78.1 | 0.0 | 0.0 |
| **$CH_3Cl$** | 538.0 | 540.8 | 457.0 | 457.0 |
| **$CH_3Br$** | 6.4 | 6.5 | 5.3 | 5.3 |
| **$CH_2Cl_2$** | 40.7 | 36.8 | 6.9 | 6.9 |



| | | | | |
|---|---|---|---|---|
| CHCl₃ | 8.7 | 8.8 | 4.8 | 4.8 |
| Halon-1211 | 3.0 | 3.3 | 0.0 | 0.0 |
| Halon-1301 | 3.4 | 3.3 | 0.0 | 0.0 |
| Halon-2402 | 0.4 | 0.4 | 0.0 | 0.0 |

## 4 Effective Radiative Forcing (ERFs)

ERFs were principally assessed in Chapter 7 of AR6 Working Group I (Forster et al., 2021). Chapter 7 focussed on assessing ERF from changes in atmospheric concentrations, it also supported estimates of ERF in Chapter 6 that attributed forcing to specific precursor emissions (Szopa et al., 2021) and also generated the time history of ERF shown in AR65 WGI Figure 2.10 and discussed in Chapter 2 (Gulev et al., 2021). Only the concentration based estimates are updated this year. The emission based estimates relied on specific chemistry climate model integrations and a consistent method of applying updates to these

would need to be developed in the future.

Each IPCC report has successively updated both the method of calculation and the time history of different warming and cooling contributions, measured as ERFs. Both types of updates have contributed to a significantly changed forcing estimate between successive reports. For example, Forster et al., (2021) updated the methodology to exclude land-surface temperature

related adjustments from the forcing calculation, which generally increased estimates. At the same time GHG levels increased and the time history of aerosol forcing was revised, overall leading to a higher total ERF estimate in AR6 compared to AR5. These IPCC updates flow from an assessment of varied literature and also rely on updates to concentrations and/or emissions.

There is no published regularly updated total ERF indicator outside of the IPCC process, although the European Copernicus

programme has trialed such a product (Bellouin et al., 2020). For radiative forcing, NOAA annually updates estimates for the main GHGs, calculating radiative forcing (RF)  using the set of formulas to estimate RFs  from concentrations (Montzka, 2022). Updated RF formulas were employed in AR6 (Forster et al., 2021) and these updated expressions are also employed here in Sect. 4.1.





The ERF calculation follows the methodology used in AR6 WGI (Smith et al., 2021b). For each category of forcing, a 100,000 member probabilistic Monte Carlo ensemble is sampled to span the assessed uncertainty range in each forcing. All uncertainties are reported as 5-95% ranges and provided in square brackets.

## 4.1 Well-mixed greenhouse gas ERF methods

Radiative forcing (RF) from $CO_2$, $CH_4$ and $N_2O$ use the simplified formulas from concentrations in Meinshausen et al. (2020),
derived from an updated functional fit to Etminan et al. (2016) line-by-line radiative transfer results. These formulas are, to first order, logarithmic with $CO_2$ concentrations and a square-root dependence for $CH_4$ and $N_2O$, with additional corrections and radiative band overlaps between gases. RF is converted to ERF using scaling factors (1.05, 0.86 and 1.07 for $CO_2$, $CH_4$, $N_2O$ respectively) that account for tropospheric and land-surface rapid adjustments (Smith et al., 2018a; Hodnebrog et al., 2020a). ERF from other GHG is assumed to scale linearly with their concentration based on their radiative efficiencies
expressed in W m$^{-2}$ ppb$^{-1}$ (Hodnebrog et al., 2020b, Smith et al., 2021b). A scaling factor translating RF to ERF is implemented for CFC-11 (1.13) and CFC-12 (1.12) (Hodnebrog et al., 2020a), whereas no model evidence exists to treat ERF differently to RF for other halogenated gases.

Relative uncertainties in the ERF for $CO_2$ (± 12%), $CH_4$ (± 20%) and $N_2O$ (± 14%) are unchanged from AR6. These stem from
a combination of spectroscopic uncertainties and uncertainties in the adjustment terms converting RF to ERF; uncertainties in the volume mixing concentrations themselves are assessed to be small (Sect. 2.2). Uncertainties in the ERF from halogenated gases are treated individually and are assessed as ±19% for gases with a lifetime of 5 or more years and ±26% for shorter lifetime gases. In AR6, a ±19% uncertainty was applied to the sum of the ERF from all halogenated gases. To maintain a consistent uncertainty range across the sum of ERF from halogenated gases with AR6, we inflate the uncertainty in each
individual gas by a factor of 2.05. Uncertainties are applied by scaling the full ERF time series for each gas.

## 4.2 Aerosol ERF methods

Aerosol ERF is a combination of contributions from aerosol-radiation interactions (ERFari) and aerosol-cloud interactions (ERFaci).

### 4..2.1 Aerosol-radiation interactions

Contributions to ERFari are assumed to scale linearly with certain SLCF emissions in Sect. 2.3 ($SO_2$, BC, OC, $NH_3$, NOx and VOC) or concentrations ($CH_4$, $N_2O$ and ozone-depleting halocarbons) of primary aerosols and chemically active precursor species. The coefficients converting emissions or concentrations of each SLCF into ERF and its uncertainty come from Chapter 6 of AR6 WGI (Szopa et al., 2021), originally from CMIP6 AerChemMIP models (Thornhill et al., 2021a). We scale these




coefficients to reproduce the headline AR6 WGI ERFari assessment of -0.3 W m$^{-2}$ from 1750 to 2005-2014. Uncertainties are
applied as a scale factor for each species and applied to the whole time series.

The inclusion of more species that affect ERFari differs from the AR6 WGI calculation of ERFari in Chapter 7, which only
used SO$_2$, BC, OC and NH$_3$ (Smith et al., 2021b). In the update, these four species remain the dominant aerosol and aerosol
precursors. Additionally, these coefficients have changed slightly due to switching to CMIP6 era data:  In AR6, the coefficients
scaling emissions to ERF for SO$_2$, BC, OC and NH$_3$ were provided by CMIP5-era models (Myhre et al., 2013a). The additional
coefficients and slight changes to their magnitude had an imperceptible effect on the results but have been included to align
with current best practice. This might be important in future years as NOx and VOC precursors might make up a larger fraction
of ERFari.

**4.2.2 Aerosol-cloud interactions**

ERFaci is estimated by assuming a logarithmic relationship with the change in cloud droplet number concentration (CDNC)
as

$$\text{ERFaci} = \beta \log (1 + \Delta \text{CDNC}) \tag{1}$$

$$\Delta \text{CDNC} = s_{SO2}\Delta E_{SO2} + s_{BC}\Delta E_{BC} + s_{OC}\Delta E_{OC} \tag{2}$$


where $s_{SO2}$, $s_{BC}$ and $s_{OC}$ are sensitivities of the change in CDNC with the change in emissions of SO$_2$, BC and OC respectively
($\Delta E$). This relationship is fit to estimates of ERFaci in 13 CMIP6 models contributing results to the piClim-histaer and histSST-
piAer experiments of RFMIP and AerChemMIP, respectively, to CMIP6. The ERFaci in these 13 models is estimated using
the Approximate Partial Radiative Perturbation (APRP) method (Taylor et al., 2007; Zelinka et al., 2014).


The $s_{SO2}$, $s_{BC}$ and $s_{OC}$ values from each model are combined into a kernel density estimate and sampled 100,000 times to provide
a CMIP6-informed distribution of these parameters. To obtain $\beta$ for each sample given ($s_{SO2}$, $s_{BC}$, $s_{OC}$) a target ERFaci value for
1750 to 2005-2014 is drawn from the headline AR6 distribution of -1.0 [-1.7 to -0.3] W m$^{-2}$ and eq. (1) rearranged. This follows
a very similar procedure to AR6 and is based on Smith et al. (2021a) with three updates. Firstly, the relationships in eqs. (1)
and (2) are slightly updated and simplified. Secondly, an additional two CMIP6 models have become available since the AR6
WG1 assessment which expands the sampling pool for coefficients from 11 to 13. Thirdly, a slight error in computing ERFaci
from APRP from the CMIP6 models in Smith et al. (2021a) has been corrected (Zelinka et al., 2023).



### 4.3 Ozone ERF methods

Ozone ERF is derived from CMIP6 model based estimates. As in AR6 WGI Chapter 7, we use results from ESMs and chemical
transport models that produced historical ozone RF estimates in Skeie et al. (2020). We use only the six ESMs in Skeie et al.
(2020) that are independent, include stratospheric and tropospheric ozone chemistry, and produce observationally plausible
distributions of present-day ozone (Smith et al., 2021b). From these model time series of ozone RF from 1850 to 2014, we
infer the sensitivity of ozone RF to emissions of NOx, VOC and CO, concentrations of $CH_4$, $N_2O$ and ozone-depleting halogens,
and global mean surface temperature (GMST) anomaly. The fit of the precursor sensitivities and GMST is performed using a
least-squares curve fit, with the search bounds of each coefficient set to the 90% range (1.645 times standard deviation) of
each species' contribution to ozone forcing determined using single-forcing experiments in Thornhill et al. (2021a) from a
number of CMIP6 models contributing to AerChemMIP. UKESM1-0-LL has an anomalously large stratospheric ozone
depletion response to halocarbons (Keeble et al., 2021), so this model was excluded when constructing these ranges. In CMIP6,
experimental results that vary CO and VOC emissions separately are not available, so individual contributions from CO and
VOC to the CO+VOC total are based on their fractional contributions from ACCMIP (CMIP5-era) models in Stevenson et al.
(2013). For the global mean temperature contribution, we use the model responses to ozone forcing per degree warming in
chemistry-enabled models in abrupt-4xCO2 experiments (Thornhill et al., 2021b). Following AR6, we do not differentiate
between stratospheric and tropospheric ozone, and we also assume that ERF is the same as RF as there is limited model
evidence to suggest otherwise.

### 4.4 ERF from other anthropogenic forcers

Minor categories of anthropogenic forcers include contributions from land use and land use change other than via GHG
emissions, aviation contrails and contrail-induced cirrus, stratospheric water vapour from methane oxidation, and light
absorbing particles on snow and ice.

The methodology to estimate ERF from land use and land-use change has been updated to use a scale factor with cumulative
$CO_2$-LUC emissions since 1750. This provides a similar time history to the land use ERF in AR6 and links this directly to land
use ERF in future scenarios (Smith et al., 2021b). We anchor the 1750-2019 assessment to be the same as AR6 at -0.20 [-0.30
to -0.10] W m$^{-2}$ under this updated methodology. With this, albedo changes and effects of irrigation (mainly via low-cloud
amount) are accounted for, while other biogeophysical effects of land use and land-use change are deemed to be of second-
order importance (Smith et al., 2021b).



Stratospheric water vapour from methane oxidation was assessed to be 0.05 [0.00 to 0.10] W m⁻² in AR6 for 1750-2019. We use the same scale factor applied to methane ERF used in AR6.

The ERF from light absorbing particles on snow and ice (LAPSI) is assumed to scale with emissions of black carbon. As in AR6, the contribution from brown carbon is assumed to be negligible. We align the coefficient that converts BC emissions to ERF from LAPSI to be 0.08 [0.00 to 0.18] W m⁻² for 1750-2019.

To estimate ERF from aviation contrails and contrail-induced cirrus in AR6, emissions of NOx from the aviation sector in
CEDS were scaled to reproduce an ERF of 0.0574 [0.019 to 0.098] W m⁻² for 1750-2018 as assessed in Lee et al. (2021a). We more closely follow the original methods of Lee et al. (2021a) in this update to base our ERF estimates as closely as possible on aviation activity data. The Lee et al. (2021a) ERF time series is extended to 2019 based on aviation fuel consumption from the International Energy Agency's (IEA) World Oil Statistics (2022). For 2020, 2021 and 2022, we use fuel consumption data from the International Air Transport Association (IATA, 2022).

**4.5 Methods for estimating natural forcing**

Natural forcing is composed of solar irradiance and volcanic eruptions.

**4.5.1 Solar irradiance**

The method to compute solar forcing is unchanged from AR6, using a composite time series prepared for PMIP4 (Jungclaus et al., 2017) and CMIP6 (Matthes et al., 2017). The headline assessment of solar ERF is based on the most recent solar
minimum (2009-2019), which is unchanged from AR6. Solar ERF estimates are computed relative to complete solar cycles encompassing the full "pre-industrial" period where proxy data exists (6754 BCE to 1745 CE).

**4.5.2 Volcanic**

Volcanic ERF consists of contributions from stratospheric sulphate aerosol optical depth (sAOD; a negative forcing) and stratospheric water vapour (sWV, a positive forcing). The sAOD time series (at a nominal wavelength of 550 nm) is constructed
from a combination of four datasets which have temporal overlap. We use ice-core deposition data from HolVol v1.0 (Sigl et al., 2022) for 9500 BCE to 1900 CE. This has been extended backwards in time from the equivalent dataset used in AR6 (eVolv2k; Toohey and Sigl, 2017) which had temporal coverage of 500 BCE to 1900 CE. For 1850 to 2014 we use the CMIP6 volcanic sAOD dataset (Dhomse et al., 2020). Since 1979, the CMIP6 dataset was constructed using the Global Space-based Stratospheric Aerosol Climatology (GloSSAC) v1.0 (Thomason et al., 2018). We use an updated, extended version of
GloSSAC (v2.2) providing sAOD up to 2021, which is itself an extension of the version used in AR6 (v2.0) ending in 2018



(Kovilakam et al., 2020). The 525 nm extinction from GloSSAC is used and converted to 550 nm using an Ångstrom exponent of -2.33. For 2013 to 2022, we use the Ozone Mapping and Profiling Limb Profiler (OMPS LP) Level 3 aerosol optical depth at 745 nm, which is scaled to achieve the same time mean sAOD as GloSSAC in the overlapping 2013-2021 period as a single Ångstrom exponent is not suggested for this conversion. The 745 nm band is used as this is reported to be more stable than the

bands closer to 550 nm from OMPS LP (Taha et al., 2021). Other than for the 2013-2021 overlap between GloSSAC v2.2 and OMPS LP in which only GloSSAC is used, we use a cross-fading approach to blend datasets in overlapping periods. Differences between datasets are minimal. sAOD is converted to a radiative effect using a scaling factor of -20 as in AR6 (Smith et al., 2021b) that is representative of CMIP5 and CMIP6 models. Effective radiative forcing is calculated with reference to the change in this radiative effect since "pre-industrial", defined as the mean of all available years before 1750

CE. In other words, the mean of the pre-1750 period is defined as zero forcing.

The January 2022 eruption of Hunga Tonga-Hunga Ha'apai (HTHH) was an exceptional episode in that it emitted large amounts of water vapour into the stratosphere (Millán et al., 2022; Sellitto et al., 2022). Jenkins et al. (2023) determined the HTHH eruption increased volcanic ERF by +0.12 W m⁻² due to sWV. The 2022 volcanic ERF has therefore been increased to

account for this. sWV injections from other volcanic eruptions historically have been assumed to be negligible. This assumption for the whole Holocene is probably incorrect (1883 Krakatau may have also emitted substantial amounts of sWV (Joshi and Jones, 2009)), but at present no known proxy datasets for sWV injections from volcanic eruptions before the observational era exist. After 1991 Pinatubo there was a marked increase in sWV above Colorado (40°N) that peaked and declined over a period of around three years following the eruption (Hurst et al., 2011). However, this was significantly smaller

than the perturbation from HTHH (Millán et al., 2022), may be obscured against a background of increasing sWV from a changing QBO state (Fueglistaler and Haynes, 2005), and reanalysis data shows no obvious water vapour signal averaged across the tropical lower stratosphere (Dessler et al., 2014). We therefore do not adjust the volcanic ERF for sWV from 1991 Pinatubo or any other eruption.

**4.5 Summary of updates to effective radiative forcing**

The summary results for the anthropogenic constituents of ERF and solar irradiance in 2022 relative to 1750 are shown in Figure 2a. In Table 4 these are summarised alongside the equivalent ERFs from AR6 (1750-2019) and AR5 (1750-2011). Figure 2b shows the time evolution of ERF from 1750 to 2022.

Total anthropogenic ERF has increased to 2.91 [2.19 to 3.63] W m⁻² in 2022 relative to 1750, compared to 2.72 [1.96 to 3.48]

W m⁻² for 2019 relative to 1750 in AR6. The main contributions to this increase are from increases in greenhouse gas





concentrations and a reduction in the magnitude of aerosol forcing. Decadal trends in ERF have increased markedly and are now over 0.6 W m$^{-2}$ per decade. These are discussed further in the discussion and conclusions (Sect. 12).

The ERF from well-mixed GHGs is 3.45 [3.14 to 3.75] W m$^{-2}$ for 1750-2022, of which 2.25 W m$^{-2}$ is from $CO_2$, 0.56 W m$^{-2}$
from $CH_4$, 0.22 W m$^{-2}$ from $N_2O$ and 0.41 W m$^{-2}$ from halogenated gases. This is an increase from 3.32 [3.03 to 3.61] W m$^{-2}$ for 1750-2019 in AR6. ERFs from $CO_2$, $CH_4$ and $N_2O$ have all increased since the AR6 WG1 assessment for 1750-2019 owing to increases in atmospheric concentrations.

The total aerosol ERF (sum of ERFari and ERFaci) for 1750-2022 is -0.98 [-1.58 to -0.40] W m$^{-2}$ compared to -1.06 [-1.71 to
-0.41] W m$^{-2}$ assessed for 1750-2019 in AR6 WG1. This continues a trend of weakening aerosol forcing due to reductions in precursor emissions. The majority of this reduction is from ERFaci which is determined to be -0.77 [-1.33 to -0.23] W m$^{-2}$ compared to -0.84 [-1.45 to -0.25] W m$^{-2}$ in AR6 for 1750-2019. ERFari for 1750-2022 is -0.21 [-0.42 to 0.00] W m$^{-2}$, marginally weaker than the -0.22 [-0.47 to 0.04] W m$^{-2}$ assessed for 1750-2019 in AR6 WG1 (Forster et al., 2021). The largest contributions to ERFari are from $SO_2$ (primary source of sulphate aerosol; -0.21 W m$^{-2}$), BC (+0.12 W m$^{-2}$), OC (-0.04 W m$^{-2}$) and $NH_3$
(primary source of nitrate aerosol; -0.03 W m$^{-2}$).

Ozone ERF is determined as 0.48 [0.24 to 0.72] W m$^{-2}$ for 1750-2022, similar to the AR6 assessment of 0.47 [0.24 to 0.71] W m$^{-2}$ for 1750-2019. Land use forcing and stratospheric water vapour from methane oxidation are unchanged (to two decimal places) since AR6. The decline in BC emissions from 2019 to 2022 has reduced ERF from light absorbing particles on snow
and ice from 0.08 [0.00 to 0.18] W m$^{-2}$ for 1750-2019 to 0.06 [0.00 to 0.14] W m$^{-2}$ for 1750-2022. We determine from provisional data that aviation activity in 2022 had not yet returned to pre-COVID levels. Therefore, ERF from contrails and contrail-induced cirrus is lower than AR6, at 0.05 [0.02 to 0.09] W m$^{-2}$ in 2022 compared to 0.06 [0.02 to 0.10] W m$^{-2}$ in 2019.

The headline assessment of solar ERF is unchanged, at 0.01 [-0.06 to +0.08] W m$^{-2}$ from pre-industrial to the 2009-2019 solar
cycle mean. Separate to the assessment of solar forcing over complete solar cycles, we provide a single year 2022 solar ERF of 0.06 [-0.02 to +0.14] W m$^{-2}$. This is higher than the single year estimate of solar ERF for 2019 (a solar minimum) of -0.02 [-0.08 to 0.06] W m$^{-2}$.

For volcanic ERF, updating of the pre-industrial dataset for sAOD from eVolv2k v3 to HolVol v1 increased the sAOD over
500 BCE to 1749 CE, resulting in a larger difference to post-1750 sAOD and resulting in a volcanic ERF difference of +0.015 W m$^{-2}$ compared to AR6. In addition, the earlier Holocene was more volcanically active than the period after 500 BCE, further increasing the mean sAOD in the pre-industrial. Taking the longer baseline period into account in HolVol, post-1750 ERF is





further increased by 0.031 W m⁻². The net effect is that volcanic forcing after 1750 has increased by +0.046 W m⁻² compared to AR6 due to dataset updates and by account of the fact that the post-1750 period was less volcanically active on average than
the early Holocene which is now used in the ERF calculation.

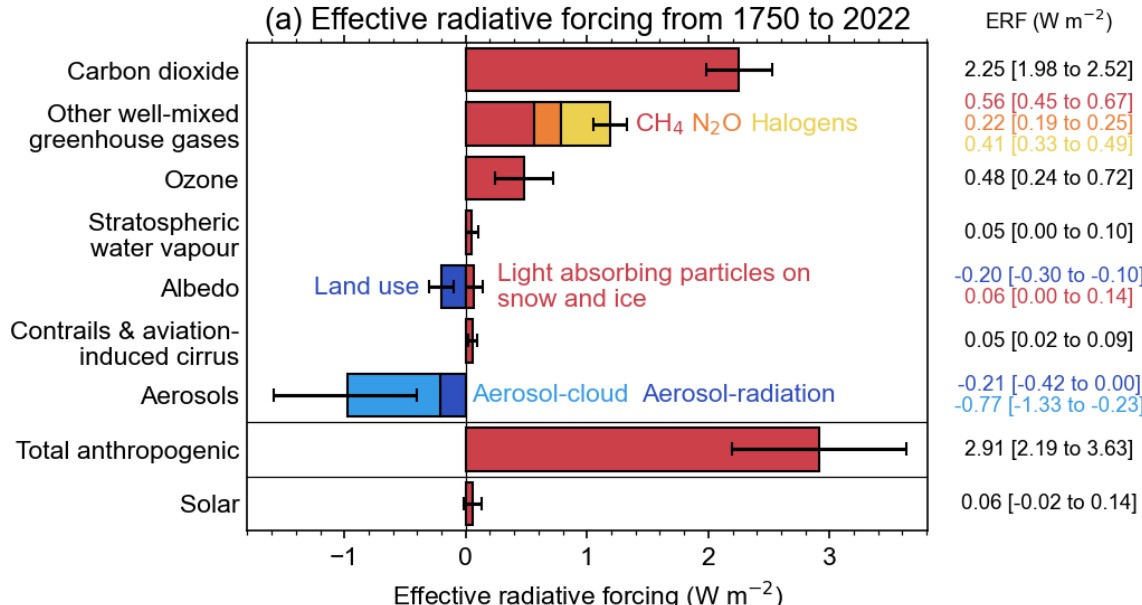

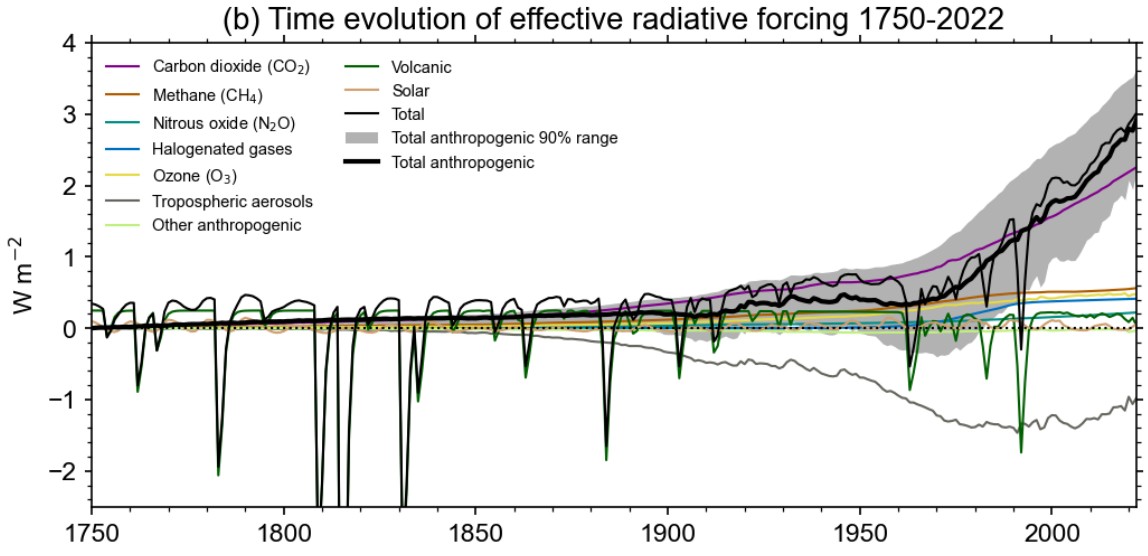





**Figure 2: Effective radiative forcing from 1750-2022. (a) 1750-2022 change in ERF, showing best estimates (bars) and 5-95% uncertainty ranges (lines) from major anthropogenic components to ERF, total anthropogenic ERF, and solar forcing. (b) Time evolution of ERF from 1750 to 2022. Best estimates from major anthropogenic categories are shown along with solar and volcanic forcing (thin coloured lines), total (thin black line) and anthropogenic total (thick black line). 5-95% uncertainty in the anthropogenic forcing is shown in shaded grey. Note solar forcing in 2022 is a single-year estimate.**

**Table 4: Contributions to anthropogenic effective radiative forcing (ERF) for 1750-2022 assessed in this section.**

| Forcer | 1750-2022 $W\,m^{-2}$ | 1750-2019 (AR6) $W\,m^{-2}$ | 1750-2011(AR5) $W\,m^{-2}$ | Reason for change from AR6 |
|---|---|---|---|---|
| $CO_2$ | 2.25 [1.98 to 2.52] | 2.16 [1.90 to 2.41] | 1.82 [1.63 to 2.01] | Increases in GHG concentrations |
| $CH_4$ | 0.56 [0.45 to 0.67] | 0.54 [0.43 to 0.65] | 0.48 [0.43 to 0.53] | |
| $N_2O$ | 0.22 [0.19 to 0.25] | 0.21 [0.18 to 0.24] | 0.17 [0.14 to 0.20] | |
| Halogenated GHGs | 0.41 [0.33 to 0.49] | 0.41 [0.33 to 0.49] | 0.36 [0.32 to 0.40] | |
| Ozone | 0.48 [0.24 to 0.72] | 0.47 [0.24 to 0.71] | 0.35 [0.21 to 0.67] | Changes in precursor emissions and chemically active GHGs; net effect almost cancels |
| Stratospheric water vapour | 0.05 [0.00 to 0.10] | 0.05 [0.00 to 0.10] | 0.07 [0.02 to 0.12] | |
| Aerosol-radiation interactions | -0.21 [-0.42 to 0.00] | -0.22 [-0.47 to 0.04] | -0.45 [-0.95 to 0.05] | Reduction in aerosol and aerosol precursor emissions |
| Aerosol-cloud interactions | -0.77 | -0.84 | -0.45 | |





|  | [-1.33 to -0.23] | [-1.45 to -0.25] | [-1.2 to 0.0] |  |
|---|---|---|---|---|
| **Land use** | **-0.20** | **-0.20** | **-0.15** |  |
|  | **[-0.30 to -0.10]** | **[-0.30 to -0.10]** | **[-0.25 to -0.05]** |  |
| **Light-absorbing particles on snow and ice** | **0.06** | **0.08** | **0.04** | **Reduction in BC emissions** |
|  | **[0.00 to 0.14]** | **[0.00 to 0.18]** | **[0.02 to 0.09]** |  |
| **Contrails and aviation-induced cirrus** | **0.05** | **0.06** | **0.05** | **As of 2022, global aviation activity has not yet returned to pre-COVID19 levels** |
|  | **[0.02 to 0.09]** | **[0.02 to 0.10]** | **[0.02 to 0.15]** |  |
| **Total anthropogenic** | **2.91** | **2.72** | **2.3** | **Increase in GHG concentrations and reduction in aerosol emissions** |
|  | **[2.19 to 3.63]** | **[1.96 to 3.48]** | **[1.1 to 3.3]** |  |
| **Solar irradiance** | **0.01** | **0.01** | **0.05** |  |
|  | **[-0.06 to 0.08]** | **[-0.06 to 0.08]** | **[0.0 to 0.10]** |  |

**All values are in W m$^{-2}$ and 5-95% ranges are in square brackets. As a comparison, the equivalent assessments from AR6 (1750-2019) and AR5 (1750-2011; Myhre et al., 2013b) are shown. Solar ERF is included and unchanged from AR6, based on the most recent solar cycle (2009-2019) thus differing from the single-year estimate in Fig. 2a. Volcanic ERF is excluded due to the sporadic nature of eruptions.**

**5. Global surface temperature**

AR6 WGI Chapter 2 assessed the 2001-2020 globally averaged surface temperature change above an 1850-1900 baseline to be 0.99 [0.84 to 1.10] °C and 1.09 [0.95 to 1.20] °C for 2011-2020 (Gulev et al., 2021). Updated estimates to 2022 were also given in AR6 SYR (Lee et al., 2023). The AR6 SYR estimates match those given here. We describe the update in detail and

provide further quantification and comparisons.

Additionally to IPCC reports, the World Meteorological Organisation (WMO) has developed a set of indicators for global climate monitoring (Trewin et al., 2020) and annually publishes 'state of the climate' reports with a global surface temperature estimate for the year of publication. For example, their 2022  "State of the Global Climate Report" gave an estimated global

surface temperature for the 2013-2022 decade of 1.14 [1.02-1.27]  °C above the 1850-1900 average (WMO, 2023). Although





the latest WMO report has worked to increase its consistency with AR6, its global surface temperature number is not directly comparable to the longer term averages given in AR6. WMO employ different datasets and incorporate reanalyses in their assessment of observed global temperature change, which AR6 did not: due to the lack of reanalysis data before 1980. The Bulletin of the American Meteorological Society (BAMS) State of the Climate report is not directly comparable as it gives
anomalies against a 1991-2020 mean and does not integrate across different datasets.

There are choices around the methods used to aggregate surface temperatures into a global average, how to correct for systematic errors in measurements, methods of infilling missing data, and whether surface measurements or atmospheric temperatures just above the surface are used. These choices, and others, affect temperature change estimates and contribute to
uncertainty (IPCC AR6 WGI Chapter 2, Cross Chapter Box 2.1, Gulev et al., 2021).

Surface temperature information on land and sea is available with low latency through WMO distribution channels, with monthly station data from a substantial number of stations reported within a few days of the end of the month. These are consolidated into global data sets by a number of institutions, making it feasible to report GMST updates within a few weeks
of the end of the period of interest. The number of reporting locations with near-real time data available for reporting for the most recent periods is typically less than that available for historical data, as not all observation sites report recent data reliably, but this lower observation density only slightly increases the uncertainty in estimates of recent annual GMST compared with the past 20-30 years Trewin et al., (2020).

The GMST assessment in AR6 was based on four datasets: HadCRUT5 (Morice et al., 2021), Berkeley Earth (Rohde and Hausfather, 2020), NOAAGlobalTemp - Interim (Vose et al., 2021) and Kadow et al. (2020). (A fifth data set, China-MST (Sun et al., 2021) was used for the land assessment only).The four GMST data sets were chosen by virtue of being quasi-globally complete, having data back to 1850, using the most recent generation of SST analyses, and using analysed (rather than climatological) values over sea ice. The first two of these are routinely updated operationally, with data for each year
becoming available in the first few weeks of the following year. NOAAGlobalTemp - Interim was not updated operationally at the time AR6 was published, but has become NOAA's main operational GMST dataset (under the name NOAAGlobalTemp 5.1) as of January 2023. All three data sets are updated and published monthly. Kadow et al. is updated on an ad hoc basis by the authors. To date, all four data sets remain supported with only minor version changes (if any) since AR6, but it is likely that more substantive version changes will occur to one or more over time, lessening direct comparability with AR6. The key
differences between the AR6 data sets and those used in the annual WMO and BAMS State of the Climate reports are that WMO and BAMS also incorporates reanalyses (ERA5 and JRA-55). These reports also include the GISTEMP (Lenssen et al.,



2019) data set (excluded by AR6 because it starts in 1880), but do not include the Kadow et al. data set yet (as that is not updated operationally).

The GMST values used in AR6 were calculated from the gridded data sets produced by the data providers, using a consistent methodology - calculating the mean anomaly for each of the northern and southern hemisphere as a latitude-weighted mean of available gridpoint values, then defining the global mean anomaly as the mean of the two hemispheric values. (This is equivalent to the method used by the Hadley Centre to report global values from HadCRUT5). The values thus calculated may differ from those reported by the data providers themselves, due to different averaging methodologies. Although the difference

is less pronounced in the AR6 datasets than in earlier generations of datasets, there are more gridpoints with missing data in the Southern Hemisphere than the Northern (particularly before an observation network was established on Antarctica in the 1950s), and using hemispheric means ensures that the two hemispheres are equally weighted.

The uncertainty assessment in AR6 combines the spread of the individual datasets with uncertainties derived from ensembles

for HadCRUT5 and an earlier version of NOAAGlobalTemp, with the other two datasets assumed to have the same uncertainty as HadCRUT5. HadCRUT5 is the only one of the datasets for which regularly updated ensembles are currently produced, limiting the extent to which uncertainty assessments can be regularly updated from those used in AR6. In this update it was assumed that the width of the confidence interval for each individual dataset was the same as that used in AR6.

Based on the updates available as of February 2023 (which were reported in the AR6 SYR), the change in GMST from 1850-1900 to 2013-2022, using the same underlying data sets and methodology as AR6, is 1.15 [1.00-1.25] °C, an increase of 0.06 °C within two years from the 2011-2020 value reported in the AR6 Working Group I report. The change from 1850-1900 to 2003-2022 was 1.03 [0.87-1.13] °C, 0.04 °C higher than the earlier value reported in the AR6 Working Group I report. These changes are broadly consistent with typical warming rates over the last few decades, which were assessed in AR6 as 0.76 °C

over the 1980-2020 period (using ordinary-least-square linear trends), or 0.019 °C per year (Gulev et al., 2021). They are also broadly consistent with projected warming rates from 2001-2020 to 2021-2040 reported in AR6, which are in the order of 0.025 °C per year under most scenarios (Lee et al., 2021b).

**Table 5: Estimates of surface temperature change from 1850-1900 [*very likely* ranges] for IPCC AR6 and the present study.**

| Time Period | Temperature change from 1850-1900 (°C) | |
|---|---|---|
| | IPCC AR6 | This study |



| | | |
|---|---|---|
| **Global, most recent 10 years** | **1.09 [0.95 to 1.20]** (to 2011-2020) | **1.15 [1.00 to 1.25]** (to 2013-2022) |
| **Global, most recent 20 years** | **0.99 [0.84 to 1.10]** (to 2001-2020) | **1.03 [0.87 to 1.13]** (to 2003-2022) |
| **Land, most recent 10 years** | **1.59 [1.34 to 1.83]** (to 2011-2020) | **1.65 [1.36 to 1.90]** (to 2013-2022) |
| **Ocean, most recent 10 years** | **0.88 [0.68 to 1.01]** (to 2011-2020) | **0.93 [0.73 to 1.04]** (to 2013-2022) |

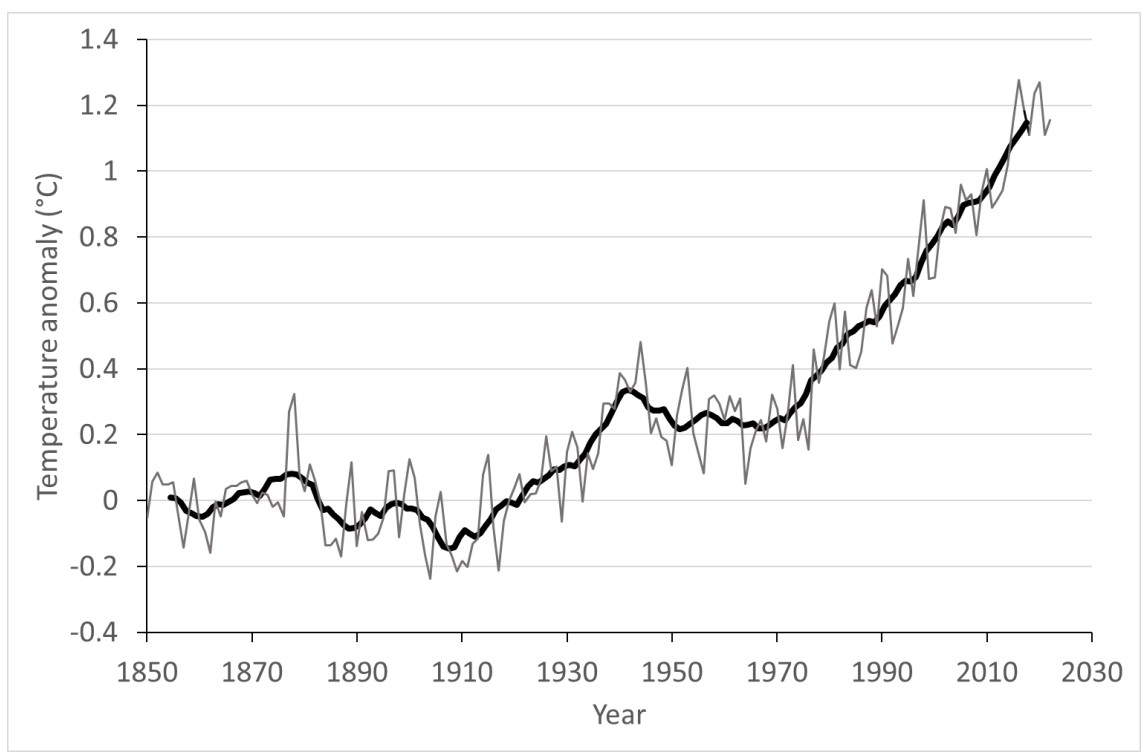


**Figure 3. Annual (thin line) and decadal (thick line) means of GMST (expressed as a change from the 1850-1900 reference period).**





Note that the temperatures for single years include considerable variability and are influenced by natural forcings such as sporadic volcanic eruptions that might either cool or warm the climate for short periods (Jenkins et al., 2023) At current
warming rates individual years may exceed warming of 1.5°C several years before a long-term mean exceeds this level (Trewin, 2022).

## 6. Earth Energy Imbalance

The Earth energy imbalance (EEI) assessed in Chapter 7 of AR6 WGI (Forster et al., 2021), provides a measure of accumulated additional energy (heating) in the climate system, and hence plays a critical role in our understanding of climate change. It
represents the difference between the radiative forcing acting to warm the climate and Earth's radiative response, which acts to oppose this warming. On annual and longer timescales, the Earth heat inventory changes associated with EEI are dominated by the changes in global ocean heat content (OHC), which accounts for about 90% of global heating since the 1970s (Forster et al, 2021). This planetary heating results in changes to the Earth system such as sea level rise, ocean warming, ice loss, rise in temperature and water vapour in the atmosphere, and permafrost thawing (e.g., von Schuckmann et al., 2023a), with adverse
impacts for ecosystems and human systems (IPCC, 2022).

On decadal timescales, changes in GMST can become decoupled from EEI by ocean heat re-arrangement processes (e.g., Palmer and McNeall, 2014; Allison et al., 2020). Therefore, the increase in the Earth heat inventory provides a more robust indicator of the rate of global change on interannual-to-decadal timescales (Forster et al., 2021; von Schuckmann et al.,
2023a). AR6 found increased confidence in the assessment of changes in the Earth heat inventory compared to previous IPCC reports due to observational advances and closure of the global sea level budget (Forster et al., 2021; Fox-Kemper et al., 2021).

AR6 estimated with *high confidence* that EEI increased from 0.50 [0.32- 0.69] W m$^{-2}$ during the period 1971-2006 to 0.79 [0.52-1.06] W m$^{-2}$ during the period 2006-2018 [*very likely* range] (Forster et al., 2021). The contributions to increases in the
Earth heat inventory throughout 1971-2018 remained fairly stable: 91% for the full-depth ocean; 5% for the land; 3% for the cryosphere and about 1% for the atmosphere (Forster et al., 2021). The increase in EEI (Figure 4) has also been reported by (von Schuckmann et al., 2020; 2023a; Loeb et al., 2021; Hakuba et al., 2021; Kramer et al., 2021; Raghuraman et al., 2021), and drivers for the most recent period (i.e., past two decades) such as cloud and sea ice changes, greenhouse gas increases, recent reductions in aerosol emissions, and planetary heat redistribution are still under discussion.

While changes in EEI have been effectively monitored at top-of-atmosphere by satellites since the mid-2000s, we rely on estimates of OHC change to determine the absolute magnitude of EEI, and its evolution on longer timescales. The AR6



assessment of ocean heat content change for the 0-2000 m layer was based on global annual mean time series from five ocean heat content datasets: IAP (Cheng et al., 2017); Domingues et al., (2008); EN4 (Good et al., 2013); Ishii et al., (2017); NCEI
(Levitus et al., 2012). Four of these datasets routinely provide updated OHC time series for the BAMS State of the Climate report, and all are used for the GCOS Earth heat inventory (von Schuckmann et al., 2020; 2023a) and the annual WMO global state of the climate. The uncertainty assessment for the 0-2000 m layer used the ensemble method described by Palmer et al. (2021) that separately accounts for *parametric* and *structural* uncertainty. The >2000 m OHC change and associated uncertainty was assessed based on trend analysis of the available hydrographic data following Purkey and Johnson (2010). All
five of the datasets used for the 0-2000 m OHC assessment are now updated at least annually and should in principle support an AR6 assessment time series update within the first few months of each year. There is potential to increase the observational ensemble used in the assessment by supplementing this set with additional data products that are also available annually for future updates.

Estimates of EEI should also account for the other elements of the Earth heat inventory, i.e., the atmospheric warming, the latent heat of global ice loss, and heating of the continental land surface (Forster et al., 2021; Cuesta-Valero et al., 2021;2022; Steiner et al., 2020; Nitzbon et al, 2022a; Vanderkelen et al., 2020; Adusumilli et al., 2022). Some of these components of the Earth heat inventory are routinely updated by a community-based initiative reported in von Schuckmann et al (2020; 2023a). However, in the absence of annual updates to all heat inventory components, a pragmatic approach is to use recent OHC
change as a proxy for EEI, scaling the value up as required based on historical partitioning between Earth system components.

We carry out an update to the AR6 estimate of changes in the Earth heat inventory based on updated observational timeseries for the period 1971-2020. Time series of heating associated with loss of ice and warming of the atmosphere and continental land surface are obtained from the recent Global Climate Observing System (GCOS) initiative (von Schuckmann et al., 2023b;
Adusumilli et al., 2022; Cuesta-Valero et al., 2023; Vanderkelen and Thiery, 2022; Nitzbon et al., 2022b; Kirchengast et al., 2022). At the time of writing, only four of the five OHC datasets used in the AR6 assessment are available with updated annual values to 2022. Therefore, our approach is to use the original AR6 time series ensemble OHC time series for the period 1971-2018 and then switch to a smaller four-member ensemble for the period 2019-2022. We "splice" the two sets of time series by adding an offset as needed to ensure that the 2018 values are identical. The AR6 heating rates and uncertainties for the ocean
below 2000 m are assumed to be time-constant. The time-evolution of the Earth heat inventory is determined as a simple summation of these time series of: atmospheric heating; continental land heating; heating of the cryosphere; and heating of the ocean over three depth layers: 0-700 m, 700-2000 m, and below 2000 m. While von Schuckmann et al (2023a) have also quantified heating of permafrost and inland lakes and reservoirs, these additional terms are very small and are omitted here for consistency with AR6 (Forster et al., 2021).






A full propagation of uncertainties across all heat inventory components is complex, and dependent on the specific choice of time-period. Therefore we take a simple pragmatic approach, using the total ocean heat content uncertainty as a proxy for the total uncertainty, since this term is two orders of magnitude larger than the other terms (Forster et al., 2021). In order to provide estimates of the EEI up to the year 2022, we scale up the values of OHC change in 2021 and 2022 to reflect the about 90%

contribution of the ocean to changes in the Earth heat inventory. The EEI is then simply computed as the difference in global energy inventory over each period, converted to units of W m⁻² using the surface area of the Earth and the elapsed time. The uncertainties in the global energy inventory for the end-point years are assumed to be independent and added in quadrature, following the approach used in AR6 (Forster et al., 2021).

**Table 6: Estimates of the Earth energy imbalance (EEI) for AR6 and the present study**

| Time Period | Earth energy imbalance (W m⁻²) Square brackets are [*very likely* ranges] | |
| --- | --- | --- |
| | IPCC AR6 | This study |
| 1971-2018 | 0.57 [0.43 to 0.72] | 0.57 [0.43 to 0.72] |
| 1971-2006 | 0.50 [0.32 to 0.69] | 0.50 [0.31 to 0.68] |
| 2006-2018 | 0.79 [0.52 to 1.06] | 0.79 [0.52 to 1.07] |
| 1975-2022 | - | 0.65 [0.48 to 0.81] |
| 2010-2022 | - | 0.89 [0.63 to 1.15] |

In our updated analysis, we find successive increases in EEI for each 20-year period since 1973, with an estimated value of 0.44 [0.05 to 0.83] W m⁻² during 1973-1992 that almost doubled to 0.82 [0.60 to 1.04] W m⁻² during 2003-2022 (Figure 4b). In the most recent decade, EEI was larger again at 0.91 [0.56 to 1.26] W m⁻² (Figure 4b rightmost bar). While there is tentative

evidence for a slight decrease in the role of the ocean in the total climate system heat uptake (95% during 1972-1992 and falling to 91% for 2011-2020), this change is unlikely to be statistically robust, given the large uncertainties. In addition, there



is some evidence that the warming signal is propagating into the deeper ocean over time, in qualitative agreement with expectations from climate model simulations (e.g. Gleckler et al, 2016). For 1973-1992 the contribution by ocean vertical layer was 66%, 28% and 1% for 0-700 m, 700-2000 m and >2000 m, respectively. During 2013-2022 the corresponding layer contributions were 50%, 33% and 8%.

The update of the AR6 assessment periods to end in 2022 results in systematic increases of EEI of 0.08 W m$^{-2}$ for 1975-2022 relative to 1971-2018 and 0.10 W m$^{-2}$ for 2010-2022 relative to 2006-2018 (Table 6).

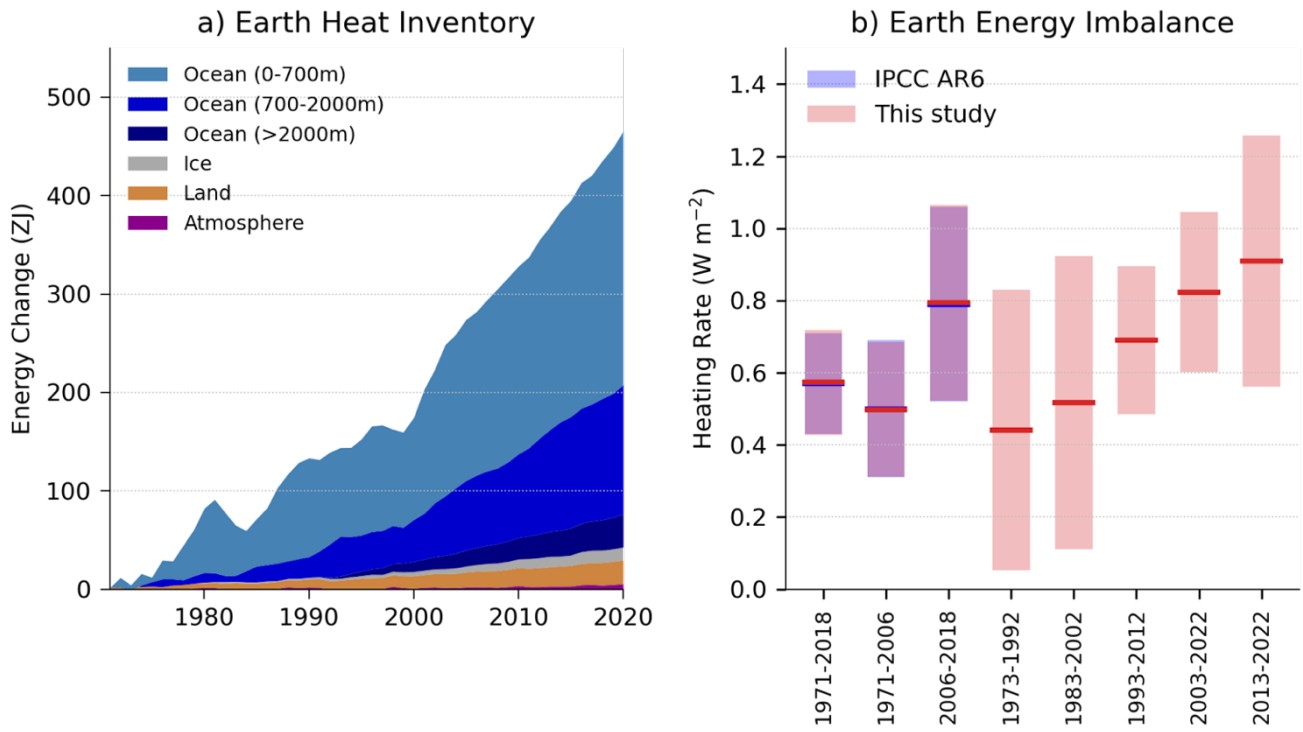

**Figure 4: a) Observed changes in the Earth heat inventory for the period 1971-2020 with component contributions as indicated in the figure legend; b) Estimates of the Earth energy imbalance for IPCC AR6 assessment periods, for consecutive twenty-year periods, and the most recent decade. Shaded regions indicate the *very likely* range (90% confidence interval). Data use and approach are based on the AR6 methods, and further described in Sect. 6.**





## 7 Human-induced global warming

### 7.1 Introduction

Human-induced warming, also known as anthropogenic warming, refers to the component of warming attributable to both the direct and indirect effects of human activities, which are typically grouped as follows: well-mixed greenhouse-gases (consisting of $CO_2$, $CH_4$, $N_2O$, and F- gases), and other-human forcings (consisting of aerosol radiation interaction, aerosol cloud interaction, black carbon on snow, contrails, ozone, stratospheric $H_2O$, and land use) (Eyring et al., 2021). While *total warming,* the actual observed temperature change resulting from both natural and anthropogenic influences, is the quantity directly related to climate impacts and therefore relevant for adaptation, mitigation efforts tend to focus on *human-induced warming* as the more relevant indicator for tracking progress towards climate targets. Further, since attributing this indicator effectively removes contributions from natural forcing and internal variability to observed warming, it can usefully minimise misperceptions about global warming arising from short-term fluctuations in temperature caused by factors such as internal variability. An assessment of human-induced warming was therefore provided in two reports within the IPCC's 6th assessment cycle: first in SR1.5 in 2018 (Chapter 1 Sect. 1.2.1.3 and Figure 1.2 (Allen et al. 2018), summarised in SPM A.1 and Figure SPM.1 (IPCC, 2018)) and second in AR6 in 2021 (WGI Chapter 3 Sect. 3.3.1.1.2 and Figure 3.8 (Eyring et al. 2021), summarised in WGI SPM A.1.3 and Figure SPM.2 (IPCC, 2021b)).

### 7.1.1 Warming Period Definitions in AR6

Each report adopted a different definition of *current* human-induced warming relative to the IPCC's 1850-1900 baseline, with AR6 defining it as the *decade-average* of the previous 10-year period (see AR6 WGI Chapter 3), and SR1.5 defining it as the average of a 30-year period centred on the *current-year* assuming the recent rate of warming continues (see SR1.5 Chapter 1). Note, if the recent rate of warming is determined by a linear trend through the most recent 15 years, this SR1.5 definition is equivalent to the *present-day* (end) value of this trendline. In practice, the absence of strong emissions reductions means that the rate of warming has been recently approximately linear, with two implications. First, the AR6 decade-average definition currently lags the *present-day single-year* value of human-induced warming by about 0.1°C. Second, the SR1.5 definition is currently almost identical to the *present-day single-year* value of human-induced warming, differing by about 0.01°C (see results in Sect. 7.4); the attribution assessment in SR1.5 was therefore provided as *single-year* warming, and could be estimated from historical data alone, assuming recent linear trends. Updates for 2022 are provided here for both the SR1.5 *present-day single-year* and AR6 *decade-average* assessment results, using methods that are directly comparable with the two IPCC definitions.



### 7.1.2 GMST and GSAT

AR6 WGI (Chapter 2 Cross-Chapter Box 2.3, Gulev et al., 2021) described how globally-complete global mean near-surface air temperature (GSAT), as is typically diagnosed from climate models, is physically distinct from the global mean surface temperature (GMST) estimated from observations, which generally combine measurements of near-surface temperature over land, and in some cases over ice, with measurements of sea surface temperature over oceans. Based on conflicting lines of evidence from climate models, which show stronger warming of GSAT compared to GMST, and observations, which tend to show the opposite, Gulev et al. (2021) assessed that long-term trends in the two indicators very likely differ by less than 10%, but that there is low confidence in the sign of the difference in trends. Therefore, with *medium* confidence, in AR6 WGI Chapter 3 (Eyring et al. (2021)), the best estimates and *likely* ranges for attributable warming expressed in terms of GMST were assessed to be equal to those for GSAT, with the consequence that the AR6 warming attribution results can be interpreted as both GMST and GSAT. While, based on the WGI Chapter 2 (Gulev et al., 2021) assessment, WGI Chapter 3 (Eyring et al. (2021)) treated estimates of attributable warming in GSAT and GMST from the literature together, without any rescaling, we note that climate-model based estimates of attributable warming in GSAT are expected to be systematically higher than corresponding estimates of attributable warming in GMST (see e.g. Cowtan et al., 2015; Richardson et al., 2018; Beusch et al., 2020; Gillett et al., 2021). Therefore, given an opportunity to update these analyses from AR6, it is more consistent, and more comparable with observations of GMST, to report attributable changes in GMST using all three methods (described in Sect. 7.2), though we also show the sensitivity of results to this choice (Appendix B). The SR1.5 assessment of attributable warming was given in terms of GMST, which is continued here. Reporting attributable warming updates in terms of GMST for both AR6 decade-average warming and SR1.5 present-day warming therefore provides improved comparability.

### 7.2 Methods

Both SR1.5 and AR6 drew on evidence from a range of literature for their assessments of human-induced warming, before selecting results from a smaller subset to produce a quantified estimate. While both the SR1.5 and AR6 assessments used the latest Global Warming Index (GWI) results (Haustein et al.,2017), AR6 also incorporated results from two other methods, Regularised Optimal Fingerprinting (ROF) (as in Gillett et al. (2021)) and Kriging for Climate Change (KCC) (as in Ribes et al. (2021)). In AR6, all three methods gave results consistent not only with each other, but also results from AR6 WGI Chapter 7 (Smith et al., 2021c), (see Figure 3.8 of AR6 WGI Chapter 3 (Eyring et al. 2021)), though the results from Chapter 7 were not included in the AR6 final assessment calculation because they were not statistically independent. Of the methods used, two (Gillett et al., 2021 and Ribes et al., 2021), relied on CMIP6 DAMIP (Gillett et al., 2016) simulations which ended in 2020, and hence require modifications to update to the most recent years. The other two methods (Haustein et al., 2017 and Smith et al., 2021c) are updatable and they can also be made consistent with other aspects of the AR6 assessment and methods.



The Gillett et al. (2021) approach can be updated by using only output from those CMIP6 models which ran individual forcing simulations beyond 2020. The three methods used in the final assessment of contributions to warming in AR6 are used again with revisions for this annual update, and outlined below.

**7.1.1 Global Warming Index**

Introduced in Otto et al. 2015, and refined with full uncertainty assessment in Haustein et al., 2017, the Global Warming Index (GWI) quantifies anthropogenic warming by using an established "multi-fingerprinting" approach to decompose total warming into its various components; preliminary anthropogenic and natural warming timeseries are first estimated from radiative forcings, and a multivariate linear regression is then taken between these preliminary GMST contributions and observed GMST, with the best-fit providing the attributed anthropogenic and natural contributions to warming. As such, the GWI attribution method is directly tied to observations, and has a low dependence on uncertainties in climate sensitivity and forcing.

Substantive annual updates to the GWI assessment depend on annual updates for effective radiative forcings (ERFs) and observed temperature (GMST), both of which are provided as a part of this update (Sects. 4 and 5 respectively). The remaining inputs to the GWI assessment are updated at the less-frequent CMIP cadence, however these contributions only weakly influence the GWI results. Further, by recomputing a "historical-only" GWI timeseries based only on data up to a given year, it can be shown that GWI is relatively insensitive to end-date or short-term fluctuations in observed GMST, minimising potential confusion about the current level of warming, such as the perception of a hiatus or acceleration (see AR6 WGI Chapter 3 Cross-Chapter Box 3.1, Eyring et al., 2021), due to short term internal variability. This, combined with the conceptual simplicity of the method, makes the GWI a relatively transparent and robust method for attributing anthropogenic warming, and well-suited to providing reliable annual updates.

Where the GWI method previously separated warming contributions into two components, 'Anthropogenic' and 'Natural', and independently attributed them, this update further separates and independently attributes contributions within the Anthropogenic component, adopting the groupings from AR6: 'Well-mixed Greenhouse Gases', 'Other Human Forcings' , and 'Natural Forcings'. The climate response model used to estimate (pre-regression) warming from radiative forcing is updated from the AR5 Impulse Response model (AR5-IR, from AR5 Chapter 8 Supplement, (Myhre et al., 2013b)) used in Haustein et al., 2017 to the Finite-amplitude Impulse Response model (FaIR, Leach et al., 2021; Smith et al., 2018b; Millar et al., 2017), which has established use in SR1.5 and AR6; climate response uncertainty is included by using around 30 sets of parameters that correspond to FaIR emulating the CMIP6 ensemble, as provided in Leach et al. (2021). The updated historical ERFs input to FaIR are given in Sect. 4, with uncertainty accounted for using a representative 1000 member probabilistic ensemble. Observed GMST and its uncertainty is provided by the 200 member ensemble of the annually updated HadCRUT5





(Morice et al. 2021, see Sect. 5). Uncertainty from internal variability is accounted for by using between 100-200 realisations

of internal variability sampled from the CMIP6 piControl simulations. Since some CMIP6 model may have unrealistically high decadal variability, implying that estimates of uncertainty may be conservative (Erying et al., 2021). Here, to partly address this, piControl timeseries are first filtered, removing simulations that drift or exhibit unrealistic variability amplitudes, changing by more than 0.15 °C per decade.

Producing the ~1 billion member GWI ensemble is computationally expensive, therefore a ~6 million member ensemble is randomly subsampled to obtain results; uncertainty converges at this scale, and repeat random samplings at the same scale lead to variation in the results of about 0.01°C.

### 7.2.2 Kriging for Climate Change

The Kriging for Climate Change method was originally introduced by Ribes et al. (2021), and subsequently extended in Qasmi

and Ribes (2022), to attribute past warming and constrain temperature projections over the 21st century. This statistical method is very similar to Ensemble Kalman Filtering, or Kriging. In the original publication (Ribes et al., 2021), a subset of 22 CMIP6 models was used to form an a priori distribution (in a Bayesian sense) of past attributable warming. Then the posterior distribution of past attributable warming given observations was derived. This application was based on HadCRUT4-CW GMST observations (Cowtan and Way, 2014), inflated by 6% to account for stronger warming of GSAT relative to GMST.

Results from this calculation were quoted in Eyring et al. (2021).

The update made here uses the same subset of 22 CMIP6 models. However, HadCRUT5 observations are used, instead of previous datasets, over an extended 1850-2022 period. Consistent with the AR6 assessment about GMST to GSAT warming ratio, no scaling correction is applied, i.e., the global mean value from HadCRUT5 is assumed to be representative for GSAT

changes (see Sect. 7.1.2). As it relies on available CMIP6 simulations, this update assumes that the world has followed a SSP2-4.5 pathway since 2015. Emissions in the SSP scenarios are similar in the period up until 2022, and close to those which have occurred (e.g. Chen et al., 2021), therefore this is a reasonable approximation. Future updates with this method will incorporate new observations. In parallel, we will try to replace the CMIP6 models by emulators, thus allowing the latest available estimates of radiative forcings to be considered, instead of the SSP2-4.5 scenario.

### 7.2.3 Regularized optimal fingerprinting

Optimal fingerprinting is the name given to optimal regression-based approaches to attribution, in which observed anomalies are regressed onto the simulated response to individual forcings from climate models, with the regression coefficients used to infer attributable contributions to observed changes (e.g. Allen and Stott, 2003; Eyring et al., 2021). Ribes et al. (2013)



proposed an improved version of the standard total least squares regression, known as regularised optimal fingerprinting, which
exhibited improved accuracy in perfect model tests. Gillett et al. (2021) applied this approach to regress observed 5-yr mean
observed GMST onto the simulated response to individual forcings from the DAMIP simulations (Gillett et al., 2016) of 13
CMIP6 models. In order to ensure a like-for-like comparison, Gillett et al. (2021) regressed observations of GMST, derived
from gridded non-infilled near-surface air temperature over land and sea ice, and sea surface temperature over oceans, onto
GMST derived from CMIP6 model output in the same way (Cowtan et al., 2015). However, since globally-complete GSAT is
usually used in the climate impacts literature which served as a basis for global warming goals, Gillett et al. (2021) used
regression coefficients to infer attributable warming in globally-complete GSAT.

Gillett et al. (2021) used CMIP6 DAMIP simulations which generally finished in 2020, and therefore cannot directly be used
to infer attributable warming in subsequent years. However, some modelling centres ran single-forcing DAMIP simulations
into the future under the SSP2-4.5 scenario (Gillett et al., 2016). Data from concatenated historical and ssp245, hist-nat and
ssp245-nat, and hist-GHG and ssp245-GHG were taken from CanESM5 (50,10,10), IPSL-CM6A-LR (11, 10, 6) and MIROC6
(3, 50, 50), where numbers in brackets indicate the respective ensemble sizes. Our approach assumes that observed drivers
have evolved as in the SSP2-4.5 scenario over the period since 2015, which is a reasonable assumption to the present (e.g.
Chen et al., 2021). As in Gillett et al. (2021), internal variability was estimated from intra-ensemble anomalies. Whereas the
Gillett et al. (2021) results assessed by Eyring et al. (2021) were based on HadCRUT4, this dataset is no longer being updated,
and therefore we use the non-infilled version of HadCRUT5 here (Morice et al., 2021). As shown by Gillett et al. (2021), using
HadCRUT5 in place of HadCRUT4 results in a 7% increase in the best-estimate of anthropogenic warming for 2010-2019.
Gillett et al. (2021) regressed 34 5-yr means of GMST over the period 1850-2019 onto simulated GMST over the same period.
Here we extend the analysis using 35 5-yr means, with the latter based on observations from January 2020 to February 2023,
and the model output masked in the same way. In order to be consistent with the Global Warming Index and Kriging for
Climate Change approaches described above, and for comparison with GMST observations, we primarily report attributable
warming in globally-complete GMST here, rather than GSAT (see Sect. 7.1.2). Finally, the use of a small ensemble of models
exposed a limitation in the approach used by Gillett et al. (2021) to account for inter-model spread in the ratio of GSAT to
GMST warming - namely that this ratio is very noisy for the naturally-forced response, due to division by numbers close to
zero sometimes occurring. Therefore we made the simplifying assumption that the contribution to the uncertainty in the GSAT
changes in the historical-nat simulations is the same as that in the historical simulations. Calculated anthropogenic warming
in GSAT in 2010-2019 computed using HadCRUT5 with this approach of 1.16 (1.04-1.29) °C (see Appendix B, Figure B1)
can be compared with the same quantity reported in Gillett et al. (2021) (Supplementary Table 1) of 1.18 (1.09-1.27) °C,
indicating good consistency.






The method described above is easily updatable into the future using the same set of simulations, simply by updating observations to a later date and masking model output accordingly. As in the KCC method, a caveat to this approach is that it relies on SSP2-4.5 simulations from which actual anthropogenic forcing might be expected to gradually diverge, and from which actual natural forcing could rapidly diverge, for example were a major volcanic eruption to occur.

**7.3 Synthesis Assessment**

**7.3.1 AR6 Assessment of Decade-Mean Attributable Warming**

Factoring in results from all three methods described above, AR6 WGI Chapter 3 (Erying et al., 2021) defined the *likely* range for each warming component as the smallest 0.1°C-precision range that enveloped the 5th to 95th percentile range of each method. In addition, a best estimate was provided for the Human-induced (Ant) warming component, calculated as the mean of the 50th percentile values for each method. Best estimates were not provided in AR6 for the other components (Well-mixed Greenhouse Gases (GHG), Other Human Forcings (OHF), Natural Forcings (Nat)), with their bars in AR6 WGI Figure SPM.2(b) simply being given as the midpoint between the lower and upper bound of the *likely* range, and therefore not directly comparable with the central bars given for human-induced and observed warming. In order to make a meaningful and consistent comparison, and provide meaningful insight into interannual changes, an improvement is made in this update: the multi-method-mean best estimate approach is extended for all warming components. The three contributing methods used in this update are the same as used in WGI AR6, with any updates to their approaches described above in Sect. 7.2.

**7.3.2 SR1.5 Assessment of Present-Day Attributable Warming**

While a variety of literature was drawn upon for the assessment of human-induced warming in SR1.5 Chapter 1 (Allen et al., 2018), only one method, the Global Warming Index (GWI), was used to provide a quantitative assessment of the 2017 level of human-induced warming. The latest results for this method were provided in Haustein et al. 2017, which gave a best estimate for human-induced warming in 2017 of 1.02°C with 5-95% range of (0.87°C to 1.22°C). SR1.5 then accounted for methodological uncertainty by rounding this value to 0.1°C precision for its final assessment of 1.0°C and assessing the 0.8°C to 1.2°C range as a *likely* range. No assessment of the contributions from other components was provided due to limitations in the GWI method at the time.

While it is possible to continue the SR1.5 assessment approach of using a single method (GWI) rounded to 0.1°C-precision, for the purpose of providing annual updates this is insufficient; (i) 0.1°C-precision is too coarse to capture meaningful inter-annual changes to the level of present-day warming, (ii) using different selections of methods prevents meaningful comparison between the results for decade-mean and present-day warming assessments, and (iii) using the mean of multiple methods





increases the robustness of the results. These points are simultaneously addressed in this update by adopting the latest multi-
method assessment approach, as established in WGI AR6, for both the WGI decade-mean warming update and the SR1.5
present-day single-year warming update. Further, where SR1.5 only provided an assessment for human-induced warming,
updates in available attribution methods since SR1.5 mean that it is now also possible to provide a fully-consistent assessment
for all warming components. As with the attribution assessment in SR1.5, this update reports values in Table 7(b) for single-

year present-day attributable warming, (as discussed in Sect. 7.1.1), with a comparison to results calculated using the SR1.5
trend based definition also provided below in Sect. 7.4.

**7.4 Results**

Results are summarised in Table 7 and Figure 5. WGI AR6 results for the period average 2010-2019 are quoted in Table 7(a),
compared with a repeat calculation using updated methods and datasets, and finally updated for the 2013-2022 period. Results

from SR1.5 are quoted in Table 7(b) for the 2017 single-year level of human-induced warming, compared with a repeat
calculation using the updated selection of methods and datasets (see Sect. 7.2) and the WGI AR6 multi-method assessment
approach (see Sect. 7.3.2), and finally updated for 2022. More details of the attributable warming timeseries and comparison
across the three methods are also given in Appendix B.

The repeat calculations for attributable warming in 2010-2019 exhibit good correspondence with the results in WGI AR6 for
the same period, (see also Appendix B, Figure B1), with an exact correspondence in the best estimate and *likely* range of
human-induced warming (Ant). The attribution assessment in WGI AR6 implied that, for the 2010-2019 decade-average,
almost all observed warming was human-induced, with natural forcings only a minor contributor; this remains true for the
2013-2022 period.


The repeat calculation for the level of attributable anthropogenic warming in 2017 is about 0.1°C larger than the estimate
provided in SR1.5; this upward correction is due to changes in observational understanding since SR1.5 (see WGI AR6 Chapter
2 Cross-Chapter Box 2.3, Table 1, Gulev et al., 2021); the attribution method (GWI) used in SR1.5 used HadCRUT4, whereas
this update uses HadCRUT5. The updated results for present-day single-year warming contributions in 2022 are therefore

higher than in 2017 due to both observational dataset updates and five additional years of anthropogenic forcing. A repeat
assessment using the SR1.5 trend-based definition leads to results that are identical to the single-year results reported in Table
7(b), except for human-induced warming being 0.02°C cooler in 2017 only, warming from natural forcings being 0.01°C
warmer in 2017 and 2022, and warming from other human forcings being cooler by 0.01°C in 2017. If warming decelerates
over the coming decade, more significant differences will arise. It is of note that the best-estimate for warming attributable to

well-mixed greenhouse gases has reached 1.49°C in 2022, offset by cooling of 0.24°C from other human forcings, with





consequences for the portfolio of multi-gas mitigation pathways available for limiting warming to 1.5°C (see Sect. 8). As with decade-mean warming, it remains true in this update that almost all present-day observed warming is attributable to anthropogenic influences.

**Table 7: Updates to assessments in the 6th IPCC assessment cycle of warming attributable to multiple influences.**

| **Estimates of warming attributable to multiple influences, in °C, relative to the 1850–1900 baseline period** Results are given as best estimates, with the *likely* range in brackets, and reported as Global Mean Surface Temperature. | | | | | | |
|---|---|---|---|---|---|---|
| Definition ➡ | **(a) IPCC AR6 Attributable Warming Update** *Average value for previous 10-year period* | | | **(b) IPCC SR1.5 Attributable Warming Update** *Present-day value for single-year period* | | |
| Period ➡ Component ⬇ | **(i) 2010-2019** *Quoted from AR6 Chapter 3 Sect. 3.3.1.1.2 Table 3.1* | **(ii) 2010-2019** *Repeat calculation using the updated methods and datasets* | **(iii) 2013-2022** *Updated value using updated methods and datasets* | **(i) 2017** *Quoted from SR1.5 Chapter 1 Sect. 1.2.1.3* | **(ii)2017** *Repeat calculation to using the updated methods and datasets* | **(iii) 2022** *Updated value using updated methods and datasets* |
| **Observed** | 1.06 (0.88 to 1.21) | 1.07 (0.89 to 1.22) * | 1.15 (1.00 to 1.25) * | | | |
| **Anthropogenic** | 1.07 (0.8 to 1.3) | 1.07 (0.8 to 1.3) | 1.14 (0.9 to 1.4) | 1.0 (0.8 to 1.2) | 1.13 (0.9 to 1.4) | 1.26 (1.0 to 1.6) |
| **Well-mixed greenhouse gases** | 1.40** (1.0 to 2.0) | 1.33 (1.0 to 1.8) | 1.40 (1.1 to 1.8) | N/A | 1.38 (1.1 to 1.8) | 1.49 (1.1 to 2.0) |
| **Other human forcings** | -0.32** (-0.8 to 0.0) | -0.26 (-0.7 to 0.1) | -0.25 (-0.7 to 0.1) | N/A | -0.25 (-0.7 to 0.1) | -0.24 (-0.7 to 0.1) |
| **Natural forcings** | 0.03** (-0.1 to 0.1) | 0.05 (-0.1 to 0.1) | 0.04 (0.0 to 0.1) | N/A | 0.04 (-0.1 to 0.2) | 0.03 (-0.1 to 0.1) |

**Results from the 6th IPCC assessment cycle, for both AR6 and SR1.5, are quoted in columns labelled (i), and are compared with repeat calculations in columns labelled (ii) for the same period using the updated methods and datasets to see how methodological and dataset updates alone would change previous assessments. Assessments for 2013-2022/2022 are reported in columns labelled (iii). Table 7.1(a): * Updated GMST observations, quoted from Sect. 5 of this update, are marked an asterisk, with very likely ranges given in brackets. ** In AR6 WGI best estimate values were not provided for warming attributable to well-mixed greenhouse gases,**
**other human forcings, and natural forcings, (though they did receive a likely range, as discussed in Sect. 7.3.1); for comparison, best estimates (marked with two asterisks) have been retrospectively calculated in an identical way to the best estimate that AR6 provided for anthropogenic warming.**



**Figure 5: Updated assessed contributions to observed warming relative to 1850-1900, following AR6 WGI SPM.2. Results for all time periods in this figure are calculated using updated datasets and methods. The 2010-2019 decade-average assessed results repeat the AR6 2010-2019 assessment, and the 2017 single-year assessed results repeat the SR1.5 2017 assessment. The 2013-2022 decade-averages and 2022 single-year results are the updated assessments for AR6 and SR1.5 respectively. Panel (a) shows updated observed global warming from Sect. 5, expressed as total GMST, due to both anthropogenic and natural influences. Whiskers give the very likely range. Panel (b) and Panel (c) show updated assessed contributions to warming, expressed as global mean surface temperature, from natural forcings and total human-induced forcings, which in turn consists of contributions from well-mixed greenhouse-gases, and other human forcings. Whiskers give the likely range.**



## 8 Remaining Carbon Budget


AR6 assessed the remaining carbon budget (RCB) in Chapter 5 of its WGI report (Canadell et al., 2021) for 1.5°C, 1.7°C and 2°C thresholds (see Table 8). They were also reported in its Summary for Policy Makers (Table SPM2, IPCC, 2021b). These are updated in this section using the same method with transparently described updates.

AR5 (IPCC, 2013) assessed that global surface temperature increase is close to linearly proportional to the total amount of cumulative $CO_2$ emissions (Collins et al., 2013). The most recent AR6 report reaffirmed this assessment (Canadell et al., 2021). This near-linear relationship implies that for keeping global warming below a specified temperature level, one can estimate the total amount of $CO_2$ that can ever be emitted, also known as the carbon budget. When expressed relative to a recent reference period, this is referred to as the remaining carbon budget (Rogelj et al., 2018).


The remaining carbon budget (RCB) is estimated by application of the SR15 method described in Rogelj et al.(2019), which involves the combination of the assessment of five factors: (i) human-induced global warming to date, (ii) the transient climate response to cumulative emissions of $CO_2$ (TCRE), (iii) the zero emissions commitment (ZEC), (iv) the temperature contribution of non-$CO_2$ emissions, and (v) an adjustment term for Earth system feedbacks that are otherwise not captured through the other factors. AR6 WGI reassessed all five terms (Canadell et al., 2021) and updated the incorporation of factor (v) (Lamboll and Rogelj, 2022).


Of these factors, only factor (i) (human-induced warming) lends itself to a regular and systematic annual update. If human-induced warming estimates are only available for decade-long periods (such as the 2010-2019 period in WGI AR6), up-to-date historical $CO_2$ emissions from the middle of this period until the start of the RCB are required to have an as up-to-date estimate as possible. However, if human-induced warming estimates are available up to the preceding year, RCB estimates can be derived without the need for estimates of historical $CO_2$ emissions.


Other factors can be updated, but depend on new evidence and insights being published rather than an additional year of observational data becoming available. Factor (iv) (temperature contribution of non-$CO_2$ emissions) depends both on the available scenario evidence and the assessment of non-$CO_2$ warming. Additional scenario evidence has become available through the publication of the AR6 WGIII report (Byers et al., 2022) which is taken into account in this update.


The RCB for 1.5°C, 1.7°C and 2°C warming levels are re-assessed based on the most recent available data. Estimated RCBs are reported below. They are expressed relative to the start of 2020 for estimates based on the 2013-2022 human-induced






warming update for comparison to WGI AR6 and relative to the start of 2023 for estimates based on the year-2022 human-induced warming update. Note that between the start of 2020 and the end of 2022, about 122 GtCO$_2$ have been emitted (Sect. 2). Based on the variation in non-CO$_2$ emissions across the scenarios in AR6 WGIII scenario database, the estimated RCB values can be higher or lower by around 200 GtCO$_2$ depending on how deeply non-CO$_2$ emissions are reduced. The impact of

non-CO$_2$ emissions on warming includes both the warming effects of other greenhouse gases such as methane and the cooling effects of aerosols such as sulphates. The impacts of these are assessed using a climate emulator (MAGICC, Meinshausen et al. 2011), which was updated to more accurately capture recent observations for the AR6 WGIII report, but whose results were not captured in the AR6 WGI carbon budget. This emulator update increased the estimate of the importance of aerosols, which are expected to decline with time, causing a net warming, and decreasing the remaining carbon budget. The AR6 WGIII version

of MAGICC is used here. If instead, the FaIR emulator were used, this would give reduced non-CO$_2$ warming and a larger carbon budget (Lamboll and Rogelj, 2022).

**Table 8: Updated estimates of the Remaining Carbon Budget for 1.5°C, 1.7°C and 2.0°C, for five levels of likelihood.**

| Case / update | Base year | Estimated remaining carbon budgets from the beginning of base year (GtCO$_2$) Likelihood of limiting global warming to temperature limit. | | | | |
|---|---|---|---|---|---|---|
| | | 17% | 33% | 50% | 67% | 83% |
| 1.5°C from AR6 WGI | 2020 | 900 | 650 | 500 | 400 | 300 |
| + AR6 emulator update | 2020 | 750 | 500 | 400 | 300 | 200 |
| + AR6 scenario update | 2020 | 750 | 500 | 400 | 300 | 200 |
| **+ warming update (2013-2022) (best estimate)** | **2023** | **500** | **300** | **250** | **150** | **100** |
| 1.7°C from AR6 WGI | 2020 | 1450 | 1050 | 850 | 700 | 550 |
| + AR6 emulator update | 2020 | 1250 | 900 | 700 | 600 | 450 |
| + AR6 scenario update | 2020 | 1300 | 950 | 750 | 600 | 500 |
| **+ warming update (2013-2022) (best estimate)** | **2023** | **1100** | **800** | **600** | **500** | **350** |
| 2°C from AR6 WGI | 2020 | 2300 | 1700 | 1350 | 1150 | 900 |
| + AR6 emulator update | 2020 | 2050 | 1500 | 1200 | 1000 | 800 |
| + AR6 WGIII scenario update | 2020 | 2200 | 1650 | 1300 | 1100 | 900 |





| + warming update (2013-2022) (best estimate) | 2023 | 2000 | 1450 | 1150 | 950 | 800 |
|---|---|---|---|---|---|---|

**Estimates start from AR6 WGI estimates (first row for each warming level), updated with the latest scenario information from AR6 WGIII (second row for each warming level), and an update of the anthropogenic historical warming which is either estimated for the 2013-2022 period (third row for each warming level). Estimates are expressed relative to the start of year 2020 or 2023 (second column). The updated estimate relative to the 2013-2022 period has been identified as the best estimate of this annual update. The probability includes only the uncertainty in how the Earth immediately responds to carbon, not long-term committed warming or uncertainty in other emissions. All values are rounded to the nearest 50 GtCO$_2$.**

Updated RCB estimates presented in Table 8 for 1.5°C, 1.7°C and 2.0°C of global warming are smaller than AR6, and geophysical and other uncertainties therefore have become larger in relative terms. This is a feature that will have to be kept in mind when communicating budgets. The estimates presented here differ from those presented in the annual Global Carbon Budget (GCB) publications (Friedlingstein et al. 2022). The GCB updates have previously started from the AR6 WGI estimate and subtract the latest estimates of historical CO$_2$ emissions. The RCB estimates presented here take into account the same updates in historical CO$_2$ emissions from the GCB as well as the latest available quantification of human-induced warming to date as well as a reassessment of non-CO$_2$ warming contributions.

If the single year human-induced warming until 2022 was used directly in the RCB calculation, this would lead to similar remaining carbon budgets estimates to those from the decadal average approach used here; the 50% likelihood estimates would be unchanged although other likelihoods alter somewhat. However, we choose to only show as this was the method adopted in AR6 WGI.

The RCB for limiting warming to 1.5°C is becoming very small. It is important, however, to correctly interpret this information. RCB estimates take into account projected reductions in non-CO$_2$ emissions that are aligned with a global transition to net zero CO$_2$ emissions. This means that reductions in aerosol cooling will unmask some of the current warming in the future. Because of this feature, it is not expected that the planet will already experience 1.5°C of global warming by the time the 50% 1.5°C budget is fully exhausted.

## 9. Climate and weather extremes

Climate and weather extremes belong to the most visible human-induced climate changes. Within AR6 WGI, a full chapter was dedicated to the assessment of past and projected changes in extremes on continents (Seneviratne et al., 2021), and the chapter on ocean, cryosphere and sea level changes also provided assessments on changes in marine heatwaves (Fox-Kemper et al., 2021). Global metrics related to climate extremes include averaged changes in climate extremes, e.g., the mean increase of annual minimum and maximum temperatures on land (AR6 WGI Chapter 11, Figure 11.2, Seneviratne et al., 2021) or the





area affected by certain types of extremes (AR6 WGI Chapter 11, Box 11.1, Figure 1, Seneviratne et al., 2021; Sippel et al., 2015). In contrast to global surface temperature, extreme indicators are less established. They are therefore expected to be subject to improvements, reflecting advances in understanding. Indeed, such efforts are planned within the World Climate Research Programme (WCRP) WCRP Grand Challenge on Weather and Climate Extremes, which will likely inform the next iteration of this study.

As part of this first update, we provide an upgraded version of the analysis in Figure 11.2 from Seneviratne et al., 2021. Like the analysis of global mean temperature, the choice of data sets is based on a compromise on the length of the data record, the data availability, near-real time updates and long-term support. As the indicator (in its current form) averages over all available land grid points, the spatial coverage should be high to obtain a meaningful average, which further limits the choice of datasets. The HadEX3 dataset (Dunn et al., 2020), which is used for Figure 11.2 in Seneviratne et al. (2021), is static and does not cover years after 2018. We therefore additionally include the Berkeley Earth Surface Temperature dataset (Rohde et al. 2013), and the fifth generation ECMWF atmospheric reanalysis of the global climate (ERA5; Hersbach et al., 2020). Berkeley Earth data currently enable an analysis of annual indices up to 2021 while ERA5 is updated daily with a latency of about 5 days (and the final release occurs after 2–3 months).

Our proposed climate indicator of changes in temperature extremes consists of land-averaged annual maximum temperatures (TXx), (excluding Antarctica). For HadEX3 we select the years 1961–2018, to exclude years with insufficient data coverage, and require at least 90% temporal completeness, thus applying the same criteria as for Figure 11.2 (Seneviratne et al., 2021). Berkeley Earth provides daily maximum temperatures and we require more than 99% data availability for each individual year and grid, such that years with more than four missing days are removed. Based on this criterion, Berkeley Earth covers at least 95% of the global land area from 1955 onwards. ERA5, on the other hand, has full spatiotemporal coverage by design, and hence the entire currently available time period of 1950 to 2022 is used. The annual maximum temperature is then computed for each grid cell, and a global area-weighted average is calculated for all grid cells with at least 90% temporal completeness in the respective available period (1955–2021 and 1961–2018 for Berkeley Earth and HadEX3, while ERA5 is again not affected by this criterion). We thus enforce high data availability to adequately calculate global land-averaged TXx across all three datasets, but their coverage is not identical which introduces minor deviations in the estimated global land averages. The resulting TXx timeseries are then computed as anomalies with respect to a baseline period of 1961–1990.

To express the TXx as anomalies with respect to 1850-1900 we add an offset to all three datasets. The offset is based on the Berkeley Earth data and is derived from the linear regression of land-mean TXx to the annual mean global mean air temperature





over the period 1955 to 2020. The offset is then calculated as the slope of the linear regression times the global mean
temperature difference between the reference periods 1850-1900 and 1961-1990 (see Appendix A, Figure A1).

Our climate has warmed rapidly in the last few decades, which also manifests in changes in the occurrence and intensity of
climate and weather extremes. We visualise this with land-averaged annual maximum temperatures (TXx) from three different
datasets (ERA5, Berkeley Earth and HadEX3), expressed as anomalies with respect to the pre-industrial baseline period of
1130 1850–1900 (Figure 9). From about 1980 onwards, all employed datasets point to a strong TXx increase, which coincides with
the transition from global dimming, associated with aerosol increases, to brightening, associated with decreases (Wild et al.,
2005). Together with strongly increasing greenhouse gas emissions (Sect. 2), this explains why human-induced climate change
has emerged at an even greater pace in the last four decades than previously. For example, land-averaged annual maximum
temperatures have warmed by more than 0.5 °C in the past 10 years (1.72 °C with respect to pre-industrial conditions)
compared to the first decade of the millennium (1.22 °C; Table 9). Since the offset relative to our pre-industrial baseline period
is calculated relative to 1961–1990, within the latter period, temperature anomalies align by construction but can diverge
afterwards. In an extensive comparison of climate extreme indices across several reanalyses and observational products, Dunn
et al. (2022), point to an overall strong correspondence between temperature extreme indices across reanalysis and
observational products, with ERA5 exhibiting especially high correlations to HadEX3 among all regularly updated datasets.
This suggests that both our choice of datasets and approach to calculate anomalies does not affect our conclusion — the





intensity of heatwaves across all land areas has unequivocally increased since pre-industrial times.

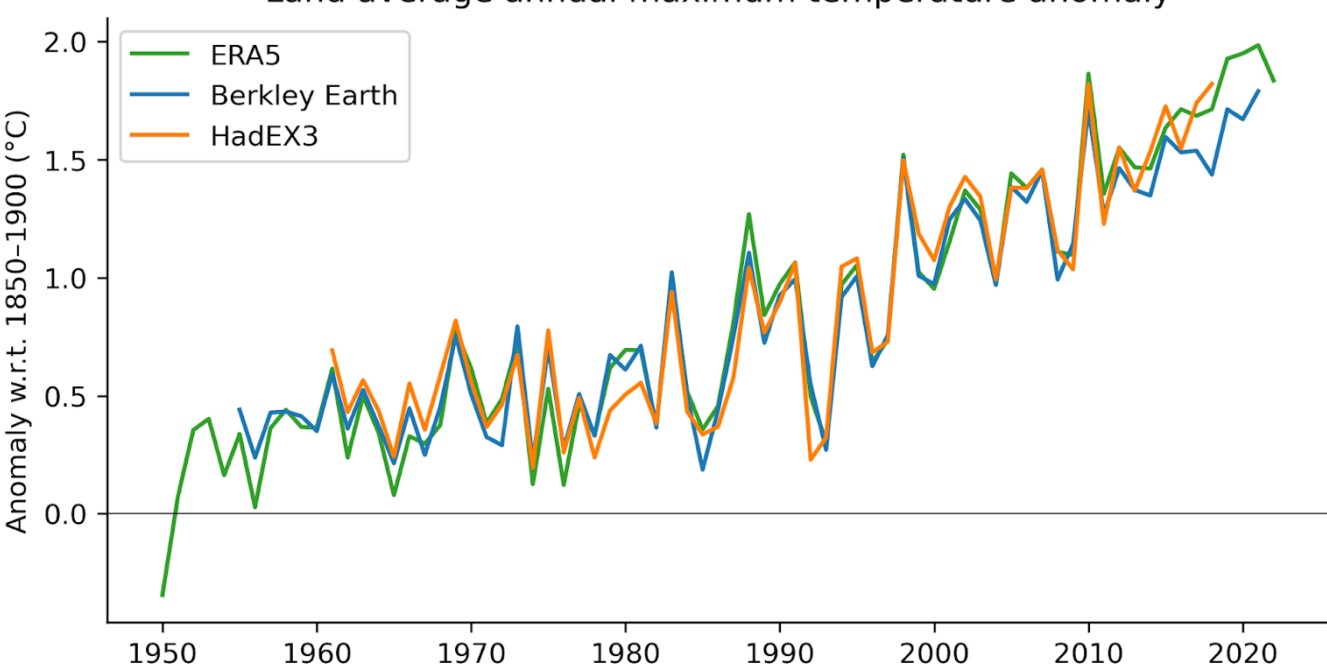

**Figure 6: Time series of observed temperature anomalies for land average annual maximum temperature (TXx) for ERA5 (1950–2022), Berkeley Earth (1955–2021), and HadEX3 (1961–2018), with respect to 1850–1900. Note that the datasets have different spatial coverage and are not coverage-matched. All anomalies are calculated relative to 1961–1990 and an offset of 0.53°C is added to obtain TXx values relative to 1850-1900. Note that while the HadEX3 numbers are the same as shown in Seneviratne et al. (2021) Figure 11.2, these numbers were not specifically assessed.**

**Table 9: Anomalies of land average annual maximum temperature (TXx) for recent decades based on HadEX3 and ERA5.**

| Period | Anomaly w.r.t. 1961-1990 (°C) | | Anomaly w.r.t. 1850-1900 (°C) |
|---|---|---|---|
| | HadEX3 | ERA5 | ERA5 |
| **Period** | | | |
| 2000-2009 | 0.72 | 0.69 | 1.23 |
| 2009-2018 | 1.01 | 1.02 | 1.55 |
| 2010-2019 | - | 1.11 | 1.64 |
| 2011-2020 | - | 1.12 | 1.65 |



| 2012-2021 | - | 1.18 | 1.71 |

**The anomalies with respect to 1850-1900 are derived by adding an offset of 0.53°C. Note that while the HadEX3 numbers are the**
**same as shown in Seneviratne et al. (2021) Figure 11.2, these numbers were not specifically assessed.**

## 10. Dashboard data visualisations

Software engineer and data scientist authors (Borger and Broersma) have created The Climate Change Tracker
(https://climatechangetracker.org/) a "dashboard" of publicly available climate data. This builds off their experience of
financial industry products: providing real time intuitive access to complex information. The aim of this tracker is to present a
range of audiences with a reliable, user-friendly platform for tracking and understanding climate change and its progression.
Building on the existing platform, we place updated IPCC-consistent indicators of climate change into the public domain via
a bespoke "dashboard" aimed primarily at policymakers involved in UNFCCC negotiations, but also intended to reach and
inform a much wider audience.

The policy-facing dashboard initially focuses on three key indicator sets: greenhouse gas emissions (Sect. 2); human-induced
global warming (Sect. 7); and the remaining global carbon budget (Sect. 8). The climate change indicator dashboard will bring
together and present up-to-date information crucial to effective climate decision-making in a findable, accessible, traceable
and reproducible way. In addition to the dashboards, the Climate Change Tracker aims to provide standardised application
programming interfaces (APIs), dashboards and charts to embed in third-party apps and websites. All data is traceable to the
sources of raw data.

In time, and with feedback from the user community, this set of indicators may be expanded to look at other indicators as well
as rates of change. However, the current aim is to engage and inform the target audience via a straightforward and easily
navigable online tool.

## 11. Code and data availability

The carbon budget calculation is available from https://github.com/Rlamboll/AR6CarbonBudgetCalc. The code and
data used to produce other indicators is available in repositories under https://github.com/ClimateIndicator. All data is
available from https://doi.org/10.5281/zenodo.7883758 (Smith et al., 2023). Data is provided under a CC-BY 4.0
Licence.



## 12. Discussion and conclusions


The first year of the Global Climate Change (IGCC) initiative has built from the AR6 report cycle to provide a comprehensive update of the climate change indicators required to estimate the human induced warming and the remaining carbon budget. Table 10 presents a summary of the headline figures from each section compared to that given in the AR6 assessment. The only substantive data change since AR6 is that land-use $CO_2$ emissions have been revised down by around 1 $GtCO_2e$. However,

as $CO_2$ ERF and human induced warming estimates depend on concentrations, not emissions, this does not affect most of the other findings. Note it does slightly increase the remaining carbon budget, but this is only by 5 $GtCO_2$, less than the 50 $GtCO_2$ rounding precision.

**Table 10, Summary of headline results and methodological updates from the Indicators of Global Climate Change (IGCC) initiative.**

| Climate Indicator | AR6 2021 assessment | This 2023 assessment | Explanation of changes | Methodological updates |
|---|---|---|---|---|
| Greenhouse gas emissions<br><br>AR6 WGIII Chapter 2: Dhakal et al. (2022); see also Minx et al. (2021) | 2010-2019 average:<br><br>56 ± 6 $GtCO_2e$* | 2012-2021 average:<br><br>57 ± 5.6 $GtCO_2e$ | Average emissions in the past decade grew at a slower rate than in the previous decade. Note following convention, ODS F-gases are excluded from the total. | Land-use emissions revised down. EDGAR historical estimates updated. These changes reduce estimates by around 1 $GtCO_2e$ (Sect. 2) |
| Greenhouse gas concentrations<br><br>AR6 WGI Chapter 2: Gulev et al. (2021) | 2019:<br><br>$CO_2$, 410.1 [± 0.36] ppm<br><br>$CH_4$, 1866.3 [± 3.2] ppb<br><br>$N_2O$, 332.1 [± 0.7] ppb | 2022:<br><br>$CO_2$, 417.1 [± 0.4] ppm<br><br>$CH_4$, 1911.9 [± 3.3] ppb<br><br>$N_2O$, 335.9 [± 0.4] ppb | Continued and increasing emissions | Updates based on NOAA data as AGAGE not yet available for 2022. To make an AR6-like product, $N_2O$ scaled to approximate NOAA-AGAGE average (Sect. 3) |
| Effective radiative forcing change since 1750<br><br>AR6 WGI Chapter 7: Forster et al. (2021) | 2019:<br><br>2.72 [1.96 to 3.48] W m$^{-2}$ | 2022:<br><br>2.91 [2.19 to 3.63] W m$^{-2}$ | Overall substantial increase and high decadal rate of change, arising from increases in greenhouse gas concentrations and reductions in aerosol precursors | Minor update in aerosol precursor method for improved future estimates - had no impact at quoted accuracy level (Sect. 4) |



| Global mean surface temperature change since preindustrial<br><br>AR6 WGI Chapter 2: Gulev et al. (2021) | 2011-2020 average:<br><br>1.09 [0.95 to 1.20] °C | 2013-2022 average:<br><br>1.15 [1.00-1.25] °C | An increase of 0.06 °C within two years, indicating a high decadal rate of change | Methods match AR6 (Sect. 5) |
|---|---|---|---|---|
| Earth's energy imbalance<br><br>AR6 WGI Chapter 7: Forster et al. (2021) | 2006-2018 average:<br><br>0.79 [0.52-1.06] W m$^{-2}$ | 2010-2022. average:<br><br>0.89 [0.63 to 1.15] W m$^{-2}$ | Substantial increase in energy imbalance estimated based on increased rate of ocean heating. | Ocean heat content timeseries extended from 2018 to 2022 using 4 of the 5 AR6 datasets. Other heat inventory terms updated following von Schuckmann et al (2023). Ocean heat content uncertainty is used as a proxy for total uncertainty. Further details in Sect. 6. |
| Human induced global warming since preindustrial<br><br>AR6 WGI Chapter 3: Eyring et al. (2021) | 2010-2019 average:<br><br>1.07 [0.8 to 1.3] °C | 2013-2023 average:<br><br>1.14 [0.9 to 1.4] °C | An increase of 0.07 °C within three years, indicating a high decadal rate of change | The three methods for the basis of the AR6 assessment are retained, but each has new input data (Sect. 7) |
| Remaining carbon budget for 50% likelihood of limiting global warming to 1.5°C<br><br>AR6 WGI Chapter 5: Canadell et al. (2021) | From the start of 2020:<br><br>500 GtCO$_2$ | From the start of 2023:<br><br>200 GtCO$_2$<br><br>(depends on stronger or weaker accompanying non-CO$_2$ mitigation) | The 1.5°C budget is becoming very small with about a 1-in-3 chance that it is already exhausted (RCB estimate for 67% is zero). The RCB will exhaust before the 1.5°C threshold is reached due to non-CO$_2$ warming that is still to be expected when CO$_2$ transitions to net zero. | Methods match AR6 (Sect. 8) |
| Land average maximum temperature change compared to preindustrial | 2009-2018 average:<br><br>1.55 °C | 2013-2022 average:<br><br>1.74 °C | Rising at a similar rate to global mean surface temperature | HadEX3 data used in AR6 replaced with reanalysis data employed in this report which is more updatable going forward as HadEX3 is static. Adds 0.01 °C to estimate (Sect. 9) |



| | | | | |
|---|---|---|---|---|
| **AR6 WGI Chapter 11: Seneviratne et al., 2021** | | | | |


Figure 7 summarises contributions to warming, repeating Figure 2.1 of the AR6 Synthesis Report (Lee et al. 2023). It highlights changes since the assessment period in ARG WGI. The table also summarises methodological updates.



**Figure 7. The causal chain from emissions to resulting warming of the climate system. Emissions of GHG have increased rapidly over recent decades (panel a). These emissions have led to increases in the atmospheric concentrations of several GHGs including the three major well-mixed GHGs (panel b). The global surface temperature (shown as annual anomalies from an 1850–1900 baseline) has increased by around 1.15°C since 1850–1900 (panel c). The human-induced warming estimate is a close match to the observed warming (panel d). Whiskers show 5% to 95% ranges. Figure is modified from AR6 SYR (Figure 2.1, Lee et al., 2023).**




It is hoped that this update can support the science community in its collection and provision of reliable and timely global climate data. In future years we are particularly interested in improving SLCF updating methods to get a more accurate estimate of short-term ERF changes. The work also highlights the importance of high-quality metadata to document changes in methodological approaches over time. In future years we hope to improve the robustness of the indicators presented here but

also extend the breadth of indicators reported through coordinated research activities. We are particularly interested in exploring how we might update indicators of regional climate extremes and their attribution, which are particularly relevant for supporting actions on adaptation and loss and damage.

Generally, scientists and scientific organisations such as WMO and IPCC have an important role as "watchdogs" to critically

inform evidence-based decision making. This annual update traced to IPCC methods can provide a reliable, timely source of trustworthy information. As well as helping inform decisions, we can use the update to track changes in dataset homogeneity between their use in one IPCC report and the next. We can also provide information and testing to motivate updates in methods that future IPCC reports might choose to employ.

Figure 8 shows decadal trends for the attributed warming and ERF. These trends were unprecedented at the time of AR6 and have increased further since then (red markers), showing that human activities are consistently causing global warming of more than 0.2 °C per decade. As nations and businesses forge climate policies and take meaningful action, our assessment shows that global actions are not yet at the scale to manifest a substantive shift in the direction of travel for global warming. Indeed, our results point to the opposite: continued high levels of greenhouse gas emissions, combined with improvements in

air quality, are reducing the level of aerosol cooling - leading to an unprecedented rate of human-induced warming. Both AR6 WGI and WGIII reports highlighted the benefits of short-term reductions in methane emissions to counter the loss of aerosol cooling and further improve air quality - however, at the global scale, methane emissions are at their highest level and rising (see Table 1). Policy makers, civil society and the scientific community require monitoring data and analyses from rigorous, robust assessments available on a regular basis. These results illustrate how assessments such as ours provide a strong "reality

check" based on science and real-world data.

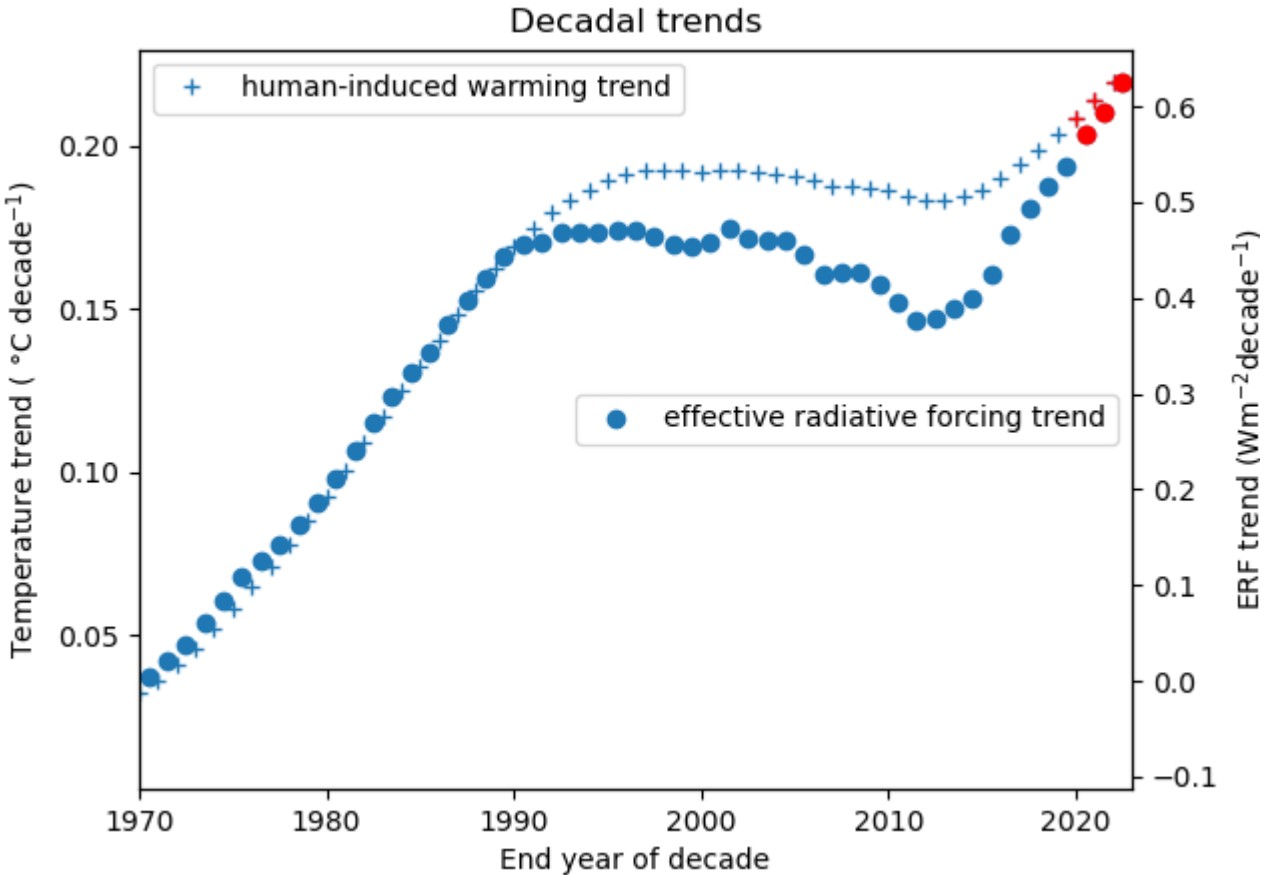

**Figure 8: Decadal trends in human-induced warming - left axis, and anthropogenic effective radiative forcing (ERF) - right axis. These are computed from the GWI human-induced warming estimate shown in Appendix B, Figure B2 and Figure 2b respectively. The red points mark three additional years since the AR6 timeseries for these indicators ended in 2019.**


This is a critical decade: warming rates are at their highest historical level and 1.5 °C global warming might be expected to be reached or passed within the next 10 years. Yet this is also the decade that global greenhouse gas emissions could be expected to peak and begin to substantially decline. The indicators of global climate change presented here show that the Earth's energy

imbalance has increased to around 0.9 W m⁻², averaged over the last 12 years. This means that there are large energy flows into the climate system and rates of human induced warming will remain high as greenhouse gas emissions remain high. Nevertheless, these warming rates do not need to be locked in as rapid emission decreases could halve warming rates over the next 20 years (McKenna et al. 2021). Table 1 shows that although global greenhouse gas emissions are at a long term high,



they are beginning to stabilise, giving some hope that over time the indicators of global climate change presented here can

track a real-world change in direction.

**Appendix A Climate and weather extremes - offset calculation**

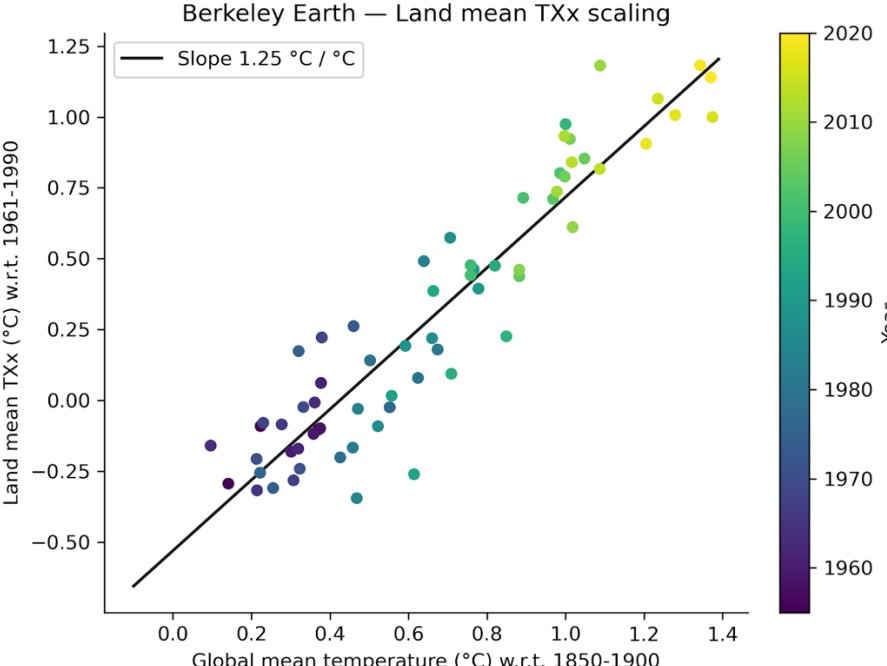

**Figure A1: Calculation of land mean annual maximum temperature (TXx) offset between 1850-1900 and 1961-1990. A linear regression of TXx as a function of global mean temperature from Berkeley Earth is fitted to data from 1955-2020. The TXx offset of 0.53 °C is then obtained by multiplying the slope of the linear regression (1.25 °C / °C) with the global mean temperature difference between 1850-1900 and 1961-1990 (0.43°C).**

## Appendix B: Human-induced global warming

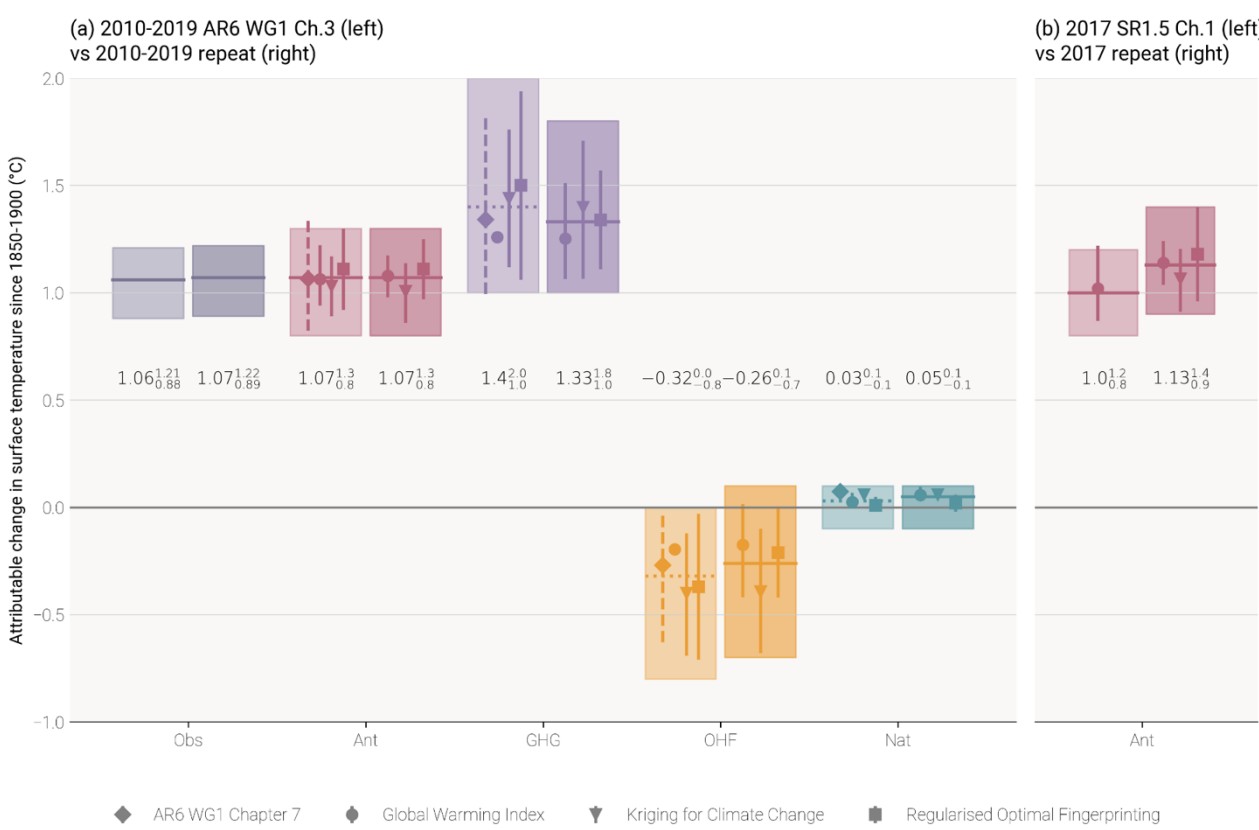

**Figure B1: Assessed contributions to observed warming and supporting lines of evidence - cf. AR6 WG1 Figure 3.8. The shaded bands show assessed likely ranges of temperature change, relative to the 1850-1900 baseline, attributable to total anthropogenic influence (Ant), well-mixed greenhouse gases (GHG), other human forcings (OHF), and natural forcings (Nat). The left of each pair of bands depicts the results quoted from AR6, and the right of each pair of bands depicts a repeat calculation for the same period as the IPCC assessment, using the revised datasets and methods, to validate the updated assessment of attributable warming. Panel (a) presents decade-average warming as used in AR6, with results quoted from AR6 WGI Chapter 3 on the left, and the repeat assessment on the right. The solid horizontal bar in each band shows the best-estimate for each warming component; if no best estimate was provided, it was retrospectively calculated using the AR6 method and depicted using a horizontal dotted line to facilitate comparison. In AR6, Global Warming Index results were reported as GMST, Kriging for Climate Change results were calculated as GMST and scaled by 1.06 for reporting as GSAT, and Regularised Optimal Fingerprinting was reported as GSAT; for the repeat, all methods are reported in terms of GMST (see Sect. 7.1.2 for discussion). Panel (b) presents single-year warming as used in SR1.5, with results quoted from SR1.5 Chapter 1 on the left (which was based only on the GWI), and the repeat assessment on the right, which now includes all of the attribution methods and the multi-method assessment approach used in AR6, as discussed in Sect. 7.3.2. Both bars are reported in GMST. No assessment was provided for components other than Ant in SR1.5.**

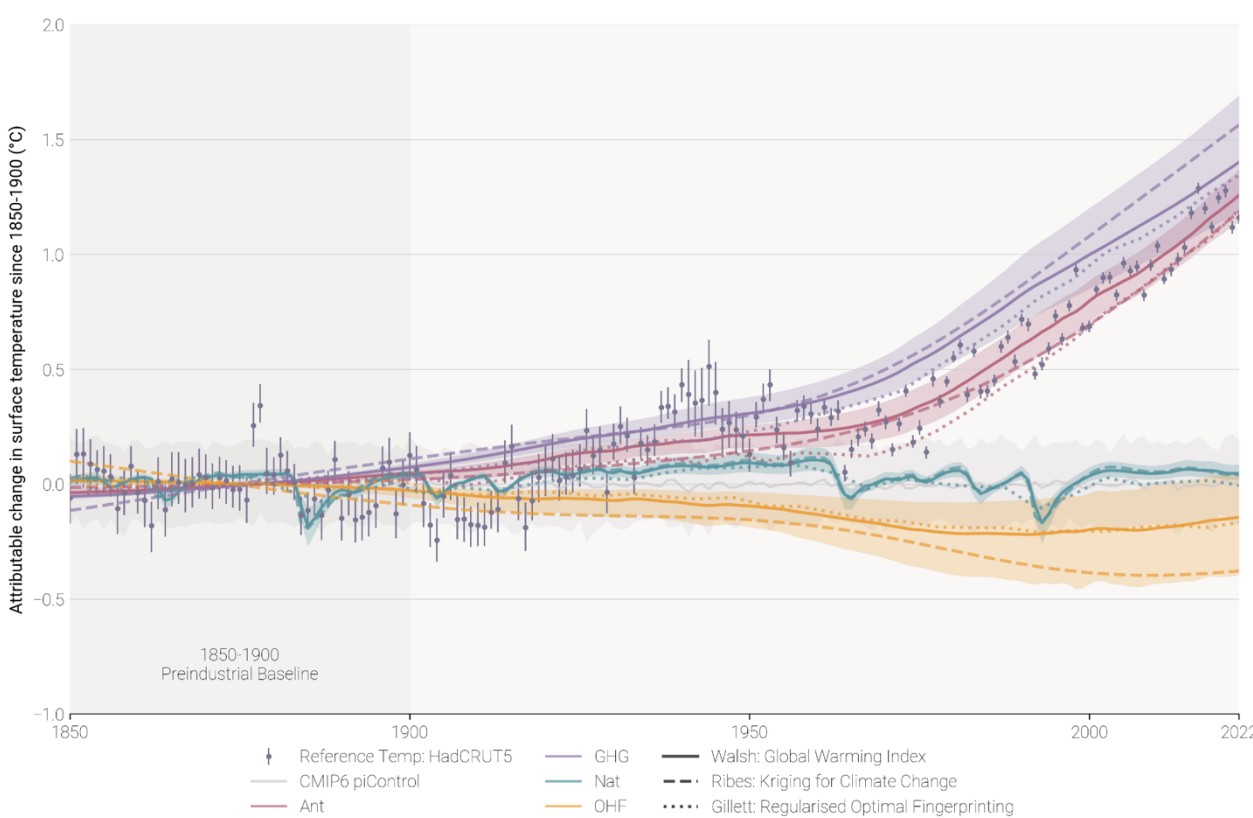


**Figure B2: Timeseries for each attribution method used in the updated assessment of warming contributions, expressed in terms of Global Mean Surface Temperature (GMST). Coloured plumes correspond to warming contributions broken down by Natural Forcings (Nat), Well-mixed Greenhouse Gases (GHG) and Other Human Forcings (OHF). Total Human-induced warming (Ant) is therefore the sum of contributions from GHG and OHF. The plume range is given by the 5-95% range of the Global Warming Index (GWI), with the GWI best estimate given by the solid lines. The dashed line presents the best estimate from the Kriging for Climate Change (KCC) method, and the dotted line presents the best estimate from the Regularised Optimal Fingerprinting (ROF) method. GWI and KCC are given as annual values based on infilled GMST from HadCRUT5; ROF is given as 5-year mean values based on non-infilled GMST from HadCRUT5 (as described in Sect. 7.2.3). The CMIP6 pre-industrial control (piControl) simulations are used as a proxy for multiple samplings of internal variability and are used to account for attribution uncertainty resulting from internal variability in the GWI method (see Sect. 7.2.1).**

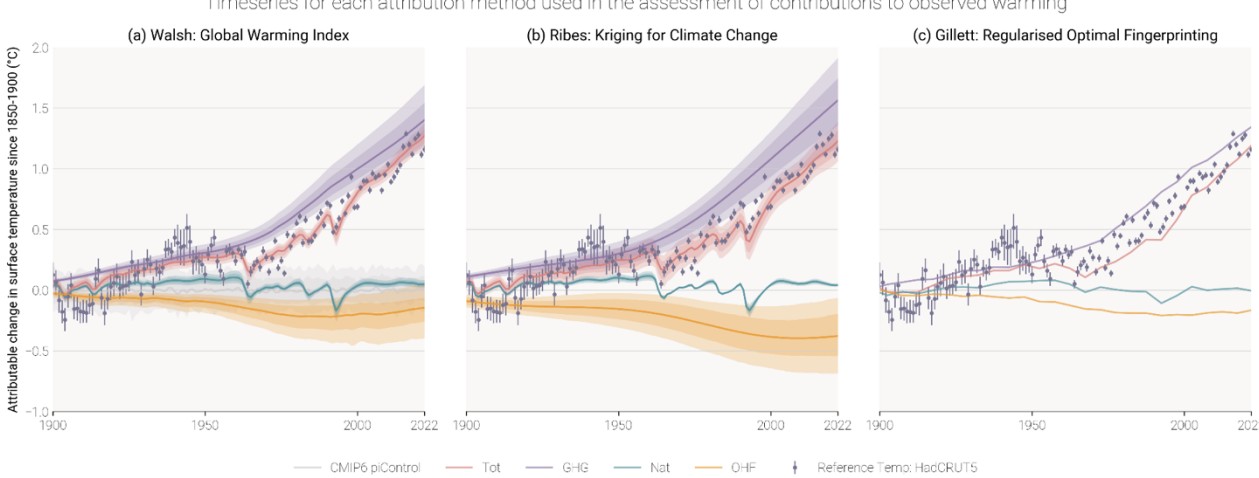

**Figure B3: Timeseries for each attribution method used in the updated assessment of warming contributions, expressed in terms of**
**Global Mean Surface Temperature (GMST). Coloured plumes are given for both 17-83% and 5-95% ranges and correspond to warming contributions to observed warming broken down by Natural Forcings (Nat), Well-mixed Greenhouse Gases (GHG) and Other Human Forcings (OHF). Total Warming (Tot) is the total attributable warming, and therefore the sum of contributions from GHG, OHF, and Nat. Observation data from (infilled) HadCRUT5 is presented with 9-95% uncertainty bars. Panel (a) presents results from the Global Warming Index method (Sect. 7.2.1); the CMIP6 pre-industrial control (piControl) simulations are used as**
**a proxy for multiple samplings of internal variability and used to account for uncertainty in the attribution resulting from internal variability (see Sect. 7.2.1). Panel (b) presents results from the Kriging for Climate Change methods (Sect. 7.2.2). Panel (c) presents results from Regularised Optimal Fingerprinting (Sect. 7.2.3), with the timeseries being given as 5-year mean values based on non-infilled GMST (as described in Sect. 7.2.3); note that this is different from GWI and KCC, which are based on infilled HadCRUT5.**

The results for each individual methods are available in csv form in the Climate Indicator repository: https://github.com/ClimateIndicator/anthropogenic-warming-assessment/.

## 13. Acknowledgements

Piers Forster, Debbie Rosen, Joeri Rogelj and Robin Lamboll were supported by the EU Horizon 2020 Research and Innovation Programme grant no.820829 (CONSTRAIN). Chris Smith was supported by a NERC/IIASA collaborative research
fellowship (NE/T009381/1). Matthew D. Palmer, Colin Morice and Rachel Killick were supported by the Met Office Hadley Centre Climate Programme funded by BEIS. William F. Lamb and Jan C. Minx were supported by the ERC-2020-SyG "GENIE" (grant ID 951542). Pierre Friedlingstein, Glen P. Peters and Robbie M. Andrew were supported by EU Horizon 2020 Research and Innovation Programme grant no. 821003 (4C). HadEX3 [3.0.4] data were obtained from https://www.metoffice.gov.uk/hadobs/hadex3/ on 05.04.2023 and are © British Crown Copyright, Met Office, 2022, provided
under an Open Government Licence http://www.nationalarchives.gov.uk/doc/open-government-licence/version/2/.



## 14.     Author contributions

PMF, CJS, MA, PF, JR, MRC and AP developed the concept of an annual update in discussions with the wider IPCC community over many years. CJS led the work of the data repositories. A. Borger and JAB led the website development with visualisation support from DR, JMG and A. Birt. VMD, PZ, SS, JM, C-FS, SIS, VN, AP, J-YL, NG, FD, GP, BT, MSP, MRC, JR, PF, MA and PT provided important IPCC and UNFCCC framing.  PMF coordinated the production of the manuscript with support from DR. WFL led Sect. 2 with contributions from CJS, JM, PF, GP, JG, JP and RA. CJS led Sects. 3 and 4 with contributions from BH, FD, SS, VN and XL. BT led Sect. 5 with contributions from PT, CM, CK, JK, RR, RV and LC. KvS and MDP led Sect. 6 with contributions from LC, MI, TB and RK. TW led Sect. 7 with contributions and calculations from AR, NG and MR. RL led Sect. 8 with contributions from JR and KZ. Sect. 9 was led by SIS and XC with calculations by MH and DS.  All authors either edited or commented on the manuscript.

## 15. Competing interests

The authors declare no competing interests.

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
