# Peer review of "Indicators of Global Climate Change 2022: Annual update of largescale indicators of the state of the climate system and the human influence"

_Earth System Science Data, 2023_

## Referee Comment (RC1)

**Review of ESSD-2023-166:**

*Indicators of Global Climate Change 2022: Annual update of large scale indicators of the state of the climate system and the human influence, by P. Forster et al.*

*Review by G. Janssens-Maenhout, 9/5/2023*

**General comment**

In line with IPCC's values to provide unbiased, traceable and transparent information and conform with the FAIR principles for sharing data, this paper documents important indicators for closely monitoring the climate change that can be attributed to human activities. While the IPCC AR (FAR until AR6) were giving similar information at relative large timesteps of 5+ years, the ESSD paper series will now continue to give for these indicators such information on an annual basis. The period covered are 10 years, 2013-2022, while AR6 covered 2010-2019. Given the acceleration in climate change and urgency for stepping up climate actions, this is most appreciated. The dataset is fully available on zenodo, authored by 13 coauthors of the ESSD paper.

However, it would have been useful to just extend the period with 3 yr before and 3 yr after the AR6 period, covering 2007-2022 for two reasons: (i) the average over the 10 yr 2010-2019 can be confronted with averages over a symmetrically extended period, (ii) 2008 saw a dip in emissions due to financial crisis and 2020 due to COVID. In the aftermath of the financial crisis, we saw a stabilising of the emissions, (Cfr. https://www.pbl.nl/en/publications/trends-in-global-co2-emissions-2013-report) but this seemed only a temporarily slow down. How can we be sure about a definite stabilising in the aftermath of COVID? It would be interesting to compare the shocks of 2008 and of 2020 and their impact on the successive years 2008-2010 en 2020-2022.

The indicators cover emissions (GHG as well as SLCF), GHG concentration, radiative forcing, surface temperature change, Earth energy imbalance, warming attributed to human activities, carbon budget and global temperature extremes. Unfortunately, indicators related to the water cycle (closely interacting with the carbon cycle) are not part of the set of indicators. While surface temperature change is a prime indicator (Fig.8 being one of the most important figures), other indicators such as soil moisture and water availability might have more direct impact on humans and nature. (Similarly, the addition of the global temperature extremes is much valued.)

Specific remarks:

Lines 78, 1234: Stabilising emission trend might be risky to claim. A slowing down of the increase might be a more prudent and correct claim.

Lines 66, 149: Emissions: please specify that these are emissions of GHGs and SLCF.

Line 200: FFI = fossil fuel and industry. It is then further explained that industrial process emissions are meant. It might be useful and clarifying to say that e.g. biomass and biofuel used in industry is not included.

Line 201: Not only deforestation but also forest degradation might be added.

Line 204: F-gases are also emitted by military activities (in the past 20% shares have been estimated). It might be clarifying to mention that these are excluded?

Line 220: there are other global inventories worth mentioning here:

- Carbon monitor https://www.nature.com/articles/s43017-023-00406-z
- TNO CAMS CO2 global emissions inventory https://coco2-project.eu/data-portal

Line 232 and line 260: the uncertainty of CO2-LULUCF is not consistent: +/- 2.6 versus +/- 2.8. Please clarify

Line 286: CH4: recent progress has been made: please take up insights from E. Nisbet et al., 2023 (DOI: 10.22541/essoar.167689502.25042797/v1) and from Z. Zhang et al., 2023 (https://www.nature.com/articles/s41558-023-01629-0)

Line 282: not only an adjustment. It seems to me that also the uncertainty got significantly reduced (from 6.6 to 5.7 Gt CO2e). Please clarify if this is also due to improvements of the LULUCF emission estimates.

Line 300: Table 1: surprisingly the projection for 2022 has the same uncertainty as the calculated value for 2021. Is the extra uncertainty for the projection so small?

Line 338: another source of SLCF emissions is Crippa et al. ESSD (2018)

Line 355: Although CEDS gives estimates for 1750, it covers also 1970. The methodology to calculate the emission estimate for 2019 is much more similar to the values in 1970 than in 1750. Why not selecting also 1970 as like for CO2e instead of 1750?

Line 400: Also here an extra column with the value for 1970 could be useful.

Line 523: when changing the dataset from IEA to IATA, is there a jump? Can this be assessed with the IATA data backwards before 2020?

Line 615, Table 4: I do not understand a trend here for the aerosol cloud interactions? Any comment on these three values?

Line 810, states to assume "recent" linear trends. I can understand that the latest increment can be linearly assessed, but is the entire trend also linearly assessed (given the multivariate linear regression method in line 850)?

Line 1025: the global surface temperature is "close" to linearly proportional to the cumulative global CO2 (not CO2e?) emissions. The causal relationship between the global surface temperature and the cumulative emissions, can be assessed with the response on a significant change. Can this be done for the change of CO2 in 2020?

Line – Table 8: It is difficult to "interpret" the different levels of likelihood to limit the global warming, I would have thought that we'll can consume our 3 yr margin (150 GtCO2e), for inducing > 1,5 deg C with 67% likelihood, but in the conclusions in Table 10, it is claimed that we exhausted already our margin and have already now a chance of 1 in 3 to exceed the 1,5 degC. How do the two Tables link?

---

## Community Comment (CC1)

**Comment on essd-2023-166: '*Indicators of Global Climate Change 2022: Annual update of large-scale indicators of the state of the climate system and the human influence*' by Forster et al.**

*Forster et al.* present a nice and worthwhile effort to keep large-scale climate indicators up to date in order to provide an up-to-date status on Earth's climate. I have a few comments about it.

Traditional sets of global climate indicators have ignored hydro-climatological aspects (e.g. the WMO, GCOS, and Copernicus), but such hydro-climatological indicators are sorely needed, for instance the global total mass of $H_2O$ falling on Earth's surface on a typical day and the fraction of Earth's surface area that receives precipitation on a typical day[1]. These may also be refined to cover land-only, oceans and different hemispheres. They can be related to both the frequency of extreme rainfall (flooding) as well as drought. One important aspect is that Earth is a closed system where the integrated moisture flux from the surface equals the total global precipitation in the long run (in a steady state).

The typically narrow emphasis on the temperature in the past is perhaps one explanation for the said incomplete set of global climate indicators (e.g. estimates of the climate sensitivity only involves temperature, but ignores the response in the global hydrological cycle to an increased greenhouse effect). L178-179 underscores this point: "*Observations of global surface temperature change (Sect. 5) and Earth's energy imbalance (Sect. 6) are key global indicators of a warming world.*" We also need to be concerned about how a strengthened greenhouse effect changes rainfall patterns and the global water budget. This aspect is often under-communicated, but extremely important (e.g. floods, droughts, water management, agriculture, health, nature).

I think that our research community should broaden out beyond the temperature focus when discussing the strengthened greenhouse effect (e.g. L790 "While total warming …"), as it is conceivable that a climate change also can involve an accelerated atmospheric overturning where increased latent heat transport keeps the temperature more in check[2]. Again, changes in
* * *
[1] Benestad, R.E., Lussana C., Lutz J., Dobler A. Landgren O.A., Haugen, J.E. Mezghani A., Casati B. Parding K.M. (2022) "Global hydro-climatological indicators and changes in the global hydrological cycle and rainfall patterns", PLOS Climate, PCLM-D-21-00079R1, DOI: 10.1371/journal.pclm.0000029

[2] Benestad, R.E. (2016) A mental picture of the greenhouse effect: A pedagogic explanation Theoretical and Applied Climatology. May 2017, Volume 128, Issue 3–4, pp 679–688, DOI: 10.1007/s00704-016-1732-y

rainfall patterns have consequences that should not be swept under the carpet. I wonder if this aspect is too invisible, even in the IPCC reports. It is absolutely relevant for adaptation (mentioned in L792), and I think it's appropriate to acknowledge this in this paper.

In addition, there have been some issues concerning the estimation of the global mean temperature which involves an ad-hoc geographical sampling of thermometer data with a subtle effect on global trend estimates, purely due to the way there are sampled[3] - in addition to the points made in the paper about GSAT and GMST (Section 7.1.2). I recommend a greater emphasis on the *global mean sea level* (easier to explain than EEI or OHC), which is a true integrator of the heat accumulation on earth, both in terms of thermal expansion of sea water and added contribution to oceans' volume from melting land ice. A stronger emphasis on the global sea level could perhaps avoid misplaced discussions on so-called 'hiatuses'.

For the polar regions, an interesting additional index is the *fractional area with above-freezing daily temperatures*, which may be correlated (e.g. aggregated over a season or a year) with the fraction snowfall/rainfall, area of snow-cover, or area with thawing permafrost. The global fractional area may be of interest also for other observations in addition to the common area-based indicators involving sea-ice and snow-cover, e.g. cloud-cover, glaciers/ice-caps area (also their volume or numbers?), total area burned by wildfires, forest area, area with declared state of drought. In addition, I think that globally aggregated albedo, longwave radiation, and (incoming) short-wave radiation, measured from space and at the surface, provide useful indicators for the closed system that Earth represents. We are now getting more global data from satellite observations and reanalyses that enable us to look at a new set of global indicators.

There may also be opportunities to give a broader account on changes in extreme events e.g. based on the statistics of record-breaking events[4] or through a (searchable map-based with moving time windows) global catalogue of historical extreme events (tropical cyclones, tornadoes, derechos, major droughts, major floods, glacial lake outflow (GLOFs), polar lows, major mid-latitude cyclones, atmospheric rivers, etc.).

In summary, I recommend expanding the set of already existing indicators with especially ones describing the evolution in the global hydrological cycle (e.g. total mass of $H_2O$ falling on Earth's surface on a typical day and the fraction of the global area on which it falls). Also, I will recommend a thorough search through the literature to capture past work that is relevant - sometimes I get the impression that we are lazy and only cite works from friends and peers from our close circles (it would be nice if we could ask ChatGPT to suggest relevant work and
* * *
[3] Benestad, R.E., Erlandsen, H.E., Parding, K.M, Mezghani, A. (2019) "Geographical distribution of thermometers gives the appearance of lower historical global warming", GRL, DOI:10.1029/2019GL083474.
[4] Benestad, R.E.(2008)  'A Simple Test for Changes in Statistical Distributions', Eos, 89 (41), 7 October 2008, p. 389-390. DOI: 10.1029/2008EO410002

references for our manuscripts). We have learned that missing out relevant work increases the risk of drawing misleading conclusions[5].
* * *
[5] Rasmus E. Benestad, Dana Nuccitelli, Stephan Lewandowsky, Katharine Hayhoe, Hans Olav Hygen, Rob van Dorland, John Cook (2015), 'Learning from mistakes in climate research', Theoretical and Applied Climatology, 126(3), 699-703, DOI:10.1007/s00704-015-1597-5

---

## Community Comment (CC8)

[supplement omitted: unrelated document]

---

## Author Response (AR1)

Reply to reviewers. We reply to reviews in turn below and indicate revised changes where appropriate. The main change is significant shortening of the main paper and adding a supplementary material. The greenhouse values have changed in light in the change to exclude GFED estimates as pointed out by reviewer 1. This required some additional rewording. Authors additionally made other minor wording changes to improve and clarify the paper. The authors order has also changed to highlight the effort of early career researchers better.

**Reviewer 1**

Thank you for your support for the effort, and your understanding of its need and timeliness. Your review has been very helpful both for revising the framing of the paper and making corrections and clarifications resulting from your detailed comments. We have considered each comment carefully and taken most on board in our revised paper.

General comment

In line with IPCC's values to provide unbiased, traceable and transparent information and conform with the FAIR principles for sharing data, this paper documents important indicators for closely monitoring the climate change that can be attributed to human activities. While the IPCC AR (FAR until AR6) were giving similar information at relative large timesteps of 5+ years, the ESSD paper series will now continue to give for these indicators such information on an annual basis. The period covered are 10 years, 2013-2022, while AR6 covered 2010-2019. Given the acceleration in climate change and urgency for stepping up climate actions, this is most appreciated. The dataset is fully available on zenodo, authored by 13 coauthors of the ESSD paper.

Thank you for the support for the effort.

However, it would have been useful to just extend the period with 3 yr before and 3 yr after the AR6 period, covering 2007-2022 for two reasons: (i) the average over the 10 yr 2010-2019 can be confronted with averages over a symmetrically extended period, (ii) 2008 saw a dip in emissions due to financial crisis and 2020 due to COVID. In the aftermath of the financial crisis, we saw a stabilising of the emissions, (Cfr. https://www.pbl.nl/en/publications/trends-in-global-co2-emissions-2013-report) but this seemed only a temporarily slow down. How can we be sure about a definite stabilising in the aftermath of COVID? It would be interesting to compare the shocks of 2008 and of 2020 and their impact on the successive years 2008-2010 en 2020-2022.

This is an interesting idea which we discussed as an author team. Potentially it could be really useful to compare emissions and responses to financial crises. We thought though that adding extra periods to the paper in this instance would make it overly complex and also reduce the focus on making a direct comparison to AR6. The data is available though and we think this would be useful for others to follow up in detail. It should, however, be noted that the 2007-2009 and 2020-2022 periods, particularly the latter, included significant

La Niña events which would be likely to confound any assessment of the 2008/2020 shocks.

The indicators cover emissions (GHG as well as SLCF), GHG concentration, radiative forcing, surface temperature change, Earth energy imbalance, warming attributed to human activities, carbon budget and global temperature extremes. Unfortunately, indicators related to the water cycle (closely interacting with the carbon cycle) are not part of the set of indicators. While surface temperature change is a prime indicator (Fig.8 being one of the most important figures), other indicators such as soil moisture and water availability might have more direct impact on humans and nature. (Similarly, the addition of the global temperature extremes is much valued.)

Adding indicators related to the water cycle and extremes has always been the eventual aim of the author team and we very much hope to include these in future updates. The extremes of temperatures are shown to indicate where we may go in the future. Including them in this first iteration was challenging as there was more work involved to establish our approach to following the IPCC processes and more subjective assessment and wider literature review required, especially to assess regional extremes. We hope to have more next year though, so please watch this space! In response to your comment, we now clarify the existing scope and state our ambition in the introduction.

Specific remarks:

Lines 78, 1234: Stabilising emission trend might be risky to claim. A slowing down of the increase might be a more prudent and correct claim.

We agree and have modified the wording using your suggestion, thank you.

Lines 66, 149: Emissions: please specify that these are emissions of GHGs and SLCF.

Both are included and we now clarify.

Line 200: FFI = fossil fuel and industry. It is then further explained that industrial process emissions are meant. It might be useful and clarifying to say that e.g. biomass and biofuel used in industry is not included.

We now clarify this as requested.

Line 201: Not only deforestation but also forest degradation might be added.

We now clarify this as requested.

Line 204: F-gases are also emitted by military activities (in the past 20% shares have been estimated). It might be clarifying to mention that these are excluded?

This paragraph is setting the scene so we think this is too much detail to add here. We now point out later in this section that the data choice we make to use atmospheric concentrations for F-gas emissions works around known issues in inventory reporting and the exclusion of military applications.

Line 220: there are other global inventories worth mentioning here:

> Carbon monitor https://www.nature.com/articles/s43017-023-00406-z
> TNO CAMS CO2 global emissions inventory https://coco2-project.eu/data-portal

These are two very useful research efforts and initiatives but they are not yet at the level of established global inventories that are cited here, so we would prefer to keep the text as is.

Line 232 and line 260: the uncertainty of CO2-LULUCF is not consistent: +/- 2.6 versus +/- 2.8. Please clarify

Line 232 refers to the uncertainty range reported by the GCB, which is calculated as a one standard deviation range. Line 260 refers to the uncertainty range reported in this study, which follows the IPCC AR6 WG3 convention of a 90% confidence interval.

Line 286: CH4: recent progress has been made: please take up insights from E. Nisbet et al., 2023 (DOI: 10.22541/essoar.167689502.25042797/v1) and from Z. Zhang et al., 2023 (https://www.nature.com/articles/s41558-023-01629-0)

We agree, and have added a short paragraph with these references.

Line 282: not only an adjustment. It seems to me that also the uncertainty got significantly reduced (from 6.6 to 5.7 Gt CO2e). Please clarify if this is also due to improvements of the LULUCF emission estimates.

Clarified, you are correct.

Line 300: Table 1: surprisingly the projection for 2022 has the same uncertainty as the calculated value for 2021. Is the extra uncertainty for the projection so small?

These are taken from the GCB estimate directly. For fossil fuel, the annual growth in emission for 2022 is 1% with a one-sigma range: 0.1 to 1.9%). Given the GCB uncertainty on the annual estimate (5%), the overall uncertainty for the GCB 2022 estimate is still around 5%. For land use emissions, the uncertainty for the 2022 GCB projection is assumed to be the same as for previous years estimates: 0.7 GtC.

Line 338: another source of SLCF emissions is Crippa et al. ESSD (2018)

This text is making a specific point on uncertainty. The 2022 EDGAR update is cited earlier.

Line 355: Although CEDS gives estimates for 1750, it covers also 1970. The methodology to calculate the emission estimate for 2019 is much more similar to the values in 1970 than in 1750. Why not selecting also 1970 as like for CO2e instead of 1750?

1750 is used specifically for the radiative forcing estimate.

Line 400: Also here an extra column with the value for 1970 could be useful.

Other periods are all in the repository, but here we wanted to focus on data periods specifically needed for the arguments in the paper. Hence the choice.

Line 523: when changing the dataset from IEA to IATA, is there a jump? Can this be assessed with the IATA data backwards before 2020?

This has now been moved to the supplement - the sources are equivalent for where they overlap.

Line 615, Table 4: I do not understand a trend here for the aerosol cloud interactions? Any comment on these three values?

Comments have been added to the table - from reduced aerosol precursors and some saturation in the system.

Line 810, states to assume "recent" linear trends. I can understand that the latest increment can be linearly assessed, but is the entire trend also linearly assessed (given the multivariate linear regression method in line 850)?

This point is about future temperatures not being needed as recent trends are roughly linear, as explained earlier in the paragraph. The multivariate linear regression is more about the scaling factors - text has been deleted to avoid confusion as it repeated the explanation given earlier but could be confusing.

Line 1025: the global surface temperature is "close" to linearly proportional to the cumulative global CO2 (not CO2e?) emissions. The causal relationship between the global

surface temperature and the cumulative emissions, can be assessed with the response on a significant change. Can this be done for the change of CO2 in 2020?

It could be, with lots of caveats around non-CO2, but this would be a research topic beyond the scope of this paper.

Line – Table 8: It is difficult to "interpret" the different levels of likehood to limit the global warming, I would have thought that we'll can consume our 3 yr margin (150 GtCO2e), for inducing > 1,5 deg C with 67% likelihood, but in the conclusions in Table 10, it is claimed that we exhausted already our margin and have already now a chance of 1 in 3 to exceed the 1,5 degC. How do the two Tables link?

Table 10 has been updated and text added. There was a legacy issue with a revision of the paper that led to Tables 10 and 8 not being consistent. Thank you for spotting this. It's to do with whether the zero emission commitment is included or excluded from the carbon budget uncertainty estimate. AR6 WGI SPM excluded it, so we follow their approach and now clarify this.

**Reviewer 2**

Thank you for your extremely helpful review of both the paper and the data repository. We have revised the paper, taking into account the comments, accepting most of them.

The authors are to be complimented for the amount of work and effort they put into this study. They have addressed most of the AR6 components and indicators. I am not commenting on the technical choice of methods, because their intention of updating AR6 is valid and greatly needed, but I find it a bit too long and descriptive, withmany detailed methodological sections. If this is targeted to scientist already familiar with the AR6 methods and GCB structure, I strongly recommend to move to a Supplementary file a lot of the methods (e.g., sections 4 (ERFs), 5 and perhaps some parts of 8, 9) and detailed tables, e.g., Table 3. I think the discussion and focus should be mostly given to section 7, where the summary of Results should have priority, perhaps merged with section 12. Now they are a bit hidden and one might get lost in methods until it arrives to read the end sections.

These are excellent suggestions for focussing the paper, thank you. We have essentially made them, additionally moving out the appendices and the three detailed methods from Section 7. We have also shortened the introduction, see response below.

After reading the Introduction, it was not completely clear to me what else was aimed at, except for updating the indicators in AR6; if the annual update of the IPCC AR6 report is the main purpose, then it might confuse the policy makers, being a scientific publication and not an official recognized IPCC report, using slightly different data sets from those in AR6; if

authors intend to also support the COP negotiations, then I see it lengthy and detailed. It also needs a more focused and concrete "bullet point" type of section including valid points for negotiations. I like the idea of having the "Reasons for change" column from Table 4 which could transform into a "summary actions for policy makers" recommendations in a final section, discussing the key figures such as: Figures 1, 2 and 7 and overview tables like Table 7.

We have refocused the introduction to remove much of the policy framing and shortened it. In response to reviewer 4 also, we want to follow the IPCC approach and keep the paper policy relevant but not policy prescriptive. We therefore do not think that adding negotiations points would be useful. Rather we want to build material around the paper to address these aspects, keeping the paper policy neutral. We are aiming to keep it as IPCC-like as possible and form conclusions in a similar way. Hopefully, the shorter and more focussed paper is more compelling.

After analyzing and updating all these indicators, what are the main messages, conclusions? Where do actions need to happen (e.g., GHG emissions, human forcing? etc.). Figure 7 is a good example.

These are brought forward in the last two paragraphs of section 12 and the abstract. We think this is the right level of messages from the paper.

The authors did not mention any indicator and/or analysis regarding the water availability, drought and sea level rise. I think it's worth mentioning it in a short section.

This is something we wish to do in future years and now cover this in the introduction when addressing the paper's scope, see also the reply to reviewer 1.

Perhaps is not the aim here, but I was wondering why satellite-based observations are still underrepresented in such studies? UNFCCC parties start to look more into complementing their NGHGIs with these estimates, e.g., CAMS or other GOSAT, TROPOMI based inversions. They do exist and provide more and more valid estimates for the last years. They could be added to Figures 1 or 5 as an extra column, or at least mentioned in the Introduction.

In this paper we stick close to AR6 methods, Bastos et al 2022 https://doi.org/10.1186/s13021-022-00214-w for example, shows that more work is still necessary to align them with bottom up methods. We now allude to future possibilities in the conclusions and will consider again for future updates.

Some specific comments:

Line 77: I would add "global emission levels are starting to stabilize…". This message sounds a bit too optimistic, and this "stabilization" is only triggered by few developed nations

managing their GHG emissions, while in most developing countries, emissions continue to increase.

We have now reworded to change the tone in line with your comments.

To keep the time lines, I would move the lines 124-134 after lines 106-113 and reduce the length of this COP21 dedicated paragraph.

These paragraphs have been shortened and removed in response to reviewer 4.

Line 201: degradation and natural disturbances

Added.

Line 204: Waste sector, important for CH4 emissions, is not mentioned.

Added.

Lines 215-220: Somewhere in Appendix or Supplement I would detail on the naming of these notable datasets of GCB. Please mention which EDGAR version is used…line 226 talks about EDGARv6.0 in AR6 while Figure 1 has EDGARv7.0

GCB uses several datasets and we think it is best for the interested reader to refer to their paper. We now mention that although EDGAR version 7 is available, we use PRIMAP.

There is a bit of confusion reading the lines 215 – 235. First the authors mention that EDGAR is used in this study (line 218), then they describe all data sets from AR6 WGIII and on line 240 they mention that they don't use EDGAR. Perhaps a simple table 'AR6 data sets vs. this study' would help summarizing the data?

The first paragraph describes the sort of data sets employed. The second one is exclusive to those used in AR6 and the third gives our updates approach. We have tried to clarify the text in the first paragraph that "not all these datasets were employed in this update." We have not added a table to keep things as simple as possible - we already have a lot of tables.

It would be good to explain why the authors consider bookkeeping models as representative for estimating the CO2-LULUCF sources/sinks? How about FAOSTAT, DGVMs, CAMS?

As specified in the text we follow AR6 and GCB standard practise. Other approaches exist that quantify land-use related CO2 fluxes, but they are not directly comparable since their definitions differ from the bookkeeping estimates as used in the annual Global Carbon Budgets, where effects of environmental changes are excluded from the LULUCF fluxes (Pongratz et al., 2021). Estimates based on DGVMs include the "loss of additional sink capacity", that is, they attribute to LULUCF the loss of the hypothetical sink that the forests,

had they not been cleared for agriculture, would have created in response to environmental changes (predominantly beneficial effects on plant growth due to rising atmospheric CO2). Their estimate of the LULUCF flux is thus a substantially higher emission term to the atmosphere (Obermeier et al., 2021). Further, by using empirical carbon densities, bookkeeping models are assumed to be more complete in their representation of management activities than the process-based DGVMs. DGVMs are thus used to quantify uncertainties around the bookkeeping estimates, but not as direct estimate of the LULUCF flux. Data sets such as the national GHG inventories by UNFCCC and FAOSTAT assume different system boundaries and distinguish fluxes by area (managed vs unmanaged) rather than by (anthropogenic vs natural) drivers. They attribute parts of the fluxes due to environmental changes to the LULUCF flux, when they occur on managed land, which thus includes a substantial sink term in the LULUCF flux estimate, turning it from a substantial global source to a global sink term. UNFCCC and bookkeeping estimates can be "translated" into each other and are largely consistent then (Friedlingstein et al., 2022; Grassi et al., 2023). Differences exist between UNFCCC and FAOSTAT in particular with respect to coverage of non-biomass carbon pools and non-forest land-use types (Grassi et al., 2022). CO2 fluxes from land can also be estimated from satellite measurements of atmospheric CO2 through inversions (Deng et al. 2022). However, currently, inversions have coarse spatial scale and country-level fluxes are relatively poorly constrained (Bastos et al. 2022). Again, the distinction between anthropogenic and natural drivers is problematic (Petrescu et al., 2021) and the key reason why models are used to estimate the LULUCF fluxes.

We have added the following text as a new paragraph in the manuscript:

"There are also varying conventions used to quantify CO2-LULUCF fluxes. These include the use of bookkeeping models, dynamic global vegetation models (DVGMs), and the national inventory approach (Pongratz et al. 2021). Each differs in terms of their applied system boundaries and definitions, and are not directly comparable. However, efforts to "translate" between bookkeeping estimates and national inventories using DVGMs have demonstrated a degree of consistency between the varying approaches (Friedlingstein et al., 2022; Grassi et al., 2023)."

This choice is discussed later in the paper, from line 245 in the submitted version.

Line 227: authors mention that in AR6 the CH4 and N2O emissions from GFED (please mention if 4.1) biomass combustion was added to EDGAR. This might create some double counting because EDGAR reports only anthropogenic emissions and emissions reported by GFED include as well agricultural waste burning and peat fires (in some countries considered managed). It is not clear to me how it's done in this study, are GFED CH4 and N2O emissions still added to the PRIMAP-hist emissions? Did you add only the wild fires?

We have investigated this important issue and found that while early drafts of the AR6 WGIII report included fire emissions from GFED, the final published draft did not. Indeed, fire emissions are an active area of development among the different teams compiling inventories - so the risk of double counting such emissions could well arise in this and subsequent editions of the indicator update. Therefore, to ensure consistency with AR6 we edit out references to GFED in the manuscript and exclude these emissions from the reported data.

You mention on line 245 the specific data choices, I do agree that higher-tier methods are needed, and I would be glad to see in the future updates the inclusion of inversions.

We will endeavor to do this.

Figure 1: please add to the caption the fact that both AR6 and current study datasets are represented.

We think the comment about the starred data used for the assessment satisfies this need and adding further text might confuse the reader.

Table 1: please correct CO2-LUCF with CO2-LULUCF

Done.

I would move Figure 4 to Appendix

We would prefer to keep it here as it clearly shows the increasing rate energy absorption by the climate system.

Figure 5: I assume all bars should have written the periods, a bit more difficult for the shorter ones. Why not adding the left and right periods on top of the bars like: a) *Decade-average warming given by observations for 2010-2019 (left) and 2013-2022 (right)* . And similar to b) and c) panels.

Only the first in the series have - this is now covered in the caption.

I find Figure 7 very informative, summarizing and concluding well the findings.

Thank you.

Zenodo data:

Please give a shorter name to the files in the "carbon_budgets" folder, error when trying to open.

Done.

The info contained by each yml file (details on the contact author and original repository of the source code) should also be added to the README file. In this way one reads the summary of data provided by the study without having to open all the file, sone by one, unless is interested in the data.

Done. Please note that rather than listing email contacts we have linked to the homepages of the relevant data contributor; while their email addresses are already mostly on their websites in plain text and contained in the YML files, we choose not to list them here in order to not provide another source for spam email crawlers.

How about individual time series from the data sets used in Section 2 (the 3 bookkeeping models, EDGAR times series, GFED, GCB? Could also be added to the greenhouse_gas_emissions_1750-2021.csv.

Thank you for the suggestion. Now, all of the emissions data has been added as an additional file in the greenhouse gas emissions dataset directory of the repository.

Please add a line in the README file: ".md and YML format files can be opened by any text editor (Notepad etc.)".

Done.

**Reviewer 3**

It's a fantastic initiative to provide running updates of all the work that underpins the IPCC assessments. This work should serve as a vital reference point for those interested in tracking (anthropogenic) climate change across a variety of indices.

The paper is extremely well written! Mercifully, the content is far more accessible than the volumous IPCC chapters that it builds on. Yet I felt that all the key concepts and background were gently spelled out, and the discussion of dataset, methods and uncertainties were covered with refreshing clarity - providing all the key information, but not so much that the text becomes convoluted. The consistent discussion of differences since AR6 and SR1.5 is also a helpful aid to the reader throughout the text.

Overall, this is a wonderful contribution to the literature that should make many aspects of future IPCC assessments far more streamlined.

Thank you so much for these very supportive comments. They make the effort involved worthwhile. We address the minor issues below and make appropriate changes where we can (some might have to wait until next time).

I have only some minor comments/suggestions:

[75] "Human induced warming is increasing at an unprecedented rate of over 0.2 °C per decade." - is this over the past decade, or extrpolated from the 2022 value? (I imagine the former, but it's not clear when reading on from the following sentence, so it would be good to specify).

"Over the 2013-2022 period" has been added to the abstract.

[Table 1] It might be nice to split the CH4 and N2O emissions from into their FFI and LUCF components, like for CO2? I think this is possible and straightforward when using the PRIMAP-hist dataset.

To avoid complexity and to follow the AR6 approach, we decided not to do this as this probably would require a fuller assessment to do properly. We will revisit it next year.

[Tables 1-3] Could you report the values for consistent periods/years across these tables? e.g. 1750, 1850, 1970-1979, ... 2010-2019, 2019, 2021, 2022.

[All results tables] As an extension of the comment above, could you provide values consistently across all results tables? I think this would measurably help the reader to digest the array of indicators provided, without getting too caught up with tracking different time periods and so on. I imagine this would help a lot with communicating the results beyond the paper - e.g. in a lecture setting.

We completely agree and set out to do just this. However, when digging into the data for the different sections, we needed to make specific date range choices to be consistent with AR6 and SR1.5. Unfortunately, these reports were not consistent across their chapters. Adding additional periods to the table to satisfy both AR6 compatibility and internal consistency would make the paper unwieldy. In this first iteration we especially wanted to show the provenance with AR6, so chose date ranges accordingly. Furthermore, in many sections of the paper such as global mean temperature, we rely on AR6 uncertainty approaches which are specific to certain averaging periods. Data for all periods are provided online, so hopefully this helps? We will revisit it again next year.

[Figure 7] Personally, I think it would be more intuitive to flip this figure along its vertical axis, so that emissions are at the top, then concentrations, then warming at the bottom. I didn't find it natural to start at the bottom and work up.

Here we choose to replicate the exact arrangement made in the AR6 synthesis report to show the provenance. This had stakeholder and designer input - so although we intend to agree, who are we to argue 🙂

[1214] "Indeed, our results point to the opposite: continued high levels of greenhouse gas emissions, combined with improvements in air quality, are reducing the level of aerosol cooling - leading to an unprecedented rate of human-induced warming". Perhaps rephrase

for clarity?: "continued high levels of greenhouse gas emissions, combined with reduced aerosol cooling under improved air quality, have led to an unprecedented rate of human-induced warming".

Your wording is adopted. Thank you.

**Reviewer 4**

This is a good and much needed paper. The idea of providing annual updates on IPCC indicators is laudable. Since most of the work relates to IPCC methods or updated versions based on published work, I have little detailed comments to make.

However, I find the introduction unnecessary cumbersome and almost too apologetic. Clearly state the aims: providing annual updates of key indicators that are changing at annual time scales and provide references to changing methods. The audience when you publish in ESDD are scientist, not policymakers

I do miss a good rationale for the present selection of indicators. A conceptual diagram, assuming the authors want to stick for now to the energy cycle,  how GHGs link to ERF, to warming, EIB might help. Now one wonders where the indicator of, for instance sea level is not used. Extremes are included, but only temperature, why not precipitation?

Thank you for this very helpful review. We agree with your issues around the framing and have shortened and focused the introduction. We have specifically reviewed much of the policy angle discussion to make it clearer that we are following IPCC of being policy relevant but policy neutral. We have strengthened the rationale over indicator choice in the introduction and explicitly explain why we do not cover precipitation extremes in this first iteration. We hope to in future updates.

Few minor details.

L92. BAMS also does GHGs

Text added to clarify.

L248 Not sure investment rate is an adequate description of the quality of a dataset. Another reason to use the UNFCC estimates, such as improved monitoring of inventories might be more useful.

"Investment" changed to "best use of country-level improvements in data gathering infrastructures".

L262. I would argue that given the uncertainty, the two values are the same, similarly l 275, there is mention of a substantial and downward revision, all very much within the uncertainty bounds, so please be more careful with the wording here. Similarly using the word "stable" is maybe a bit too positive if you compare only two years.

We agree, wording adjusted to make the text less definitive.

L400. I wonder if Table 4 is not better places in supplementary data.

Change made as requested and table moved to the Supplement - I think you meant old table 3.

Despite these comments I think the authors have done a wonderful job and I look forward seeing the indicators published in the dashboard (where I would like to make sure that the different components in the dashboard do reflect the same data and numbers as in the policy facing dashboard)

Thank you. Your help and advice for the dashboard development would be much appreciated.

**Reply to Reviewer 5**

This is a very useful update since AR6 of key climate change indicators. The authors have made a very nice job, and the manuscript reads extremely well. I have only a few comments:

Thank you for the positive endorsement, your review has been very helpful to us: thank you for taking the time.

General comments:

1) I second Han Dolman's review comment in that section 1 could be significantly shortened and focused on the key objectives of this manuscript, i.e. providing an update of key climate change indicators. The overall rationale for providing a science based assessment has been well covered in the IPCC reports.

We agree with this point and have removed the overly policy-focused paragraph to reframe the introduction. This also helps clarify the purpose of the paper.

2) Section 9, climate and weather extremes. As stated in the text, this section is a placeholder for extreme weather indicators in future updates of this effort. For now, only an update of an extreme temperature indicator is discussed. Should not the title of this section be modified to reflect this restriction? Since this section and Figure 6 provide an "update" of Figure 11.2 in AR6, one might mention here as a footnote that Figure 11.2 in the final

version of AR6 has a wrong time axis (data appear to go up to the year 2023, while the report was published in 2021…).

Good idea, thank you. We have changed the title of section 9 to "Examples of climate and weather extremes : maximum temperature over land" .We got our rulers out to confirm the last point on Figure 11.2 in AR6 Chapter 11 is for 2020, so these data are ok.

Minor comments:

p 11, Table 1: For consistency, why not have an additional column with the decade 2010-2019? I understand the reason to show the last decade, but the gap between 2009 and 2012 is somewhat puzzling.

Done.

Section 3, Table 3: I'd reverse the ordering of the columns so that time runs to the right, similar to all the other tables. An additional column with the annual increment in 2022 or perhaps averaged over 2019-2022, would also be useful.

We have reversed the order but prefer to keep single years as this builds to radiative forcing. The uncertainty in trends needs to be thought about more for minor species so this is not added this year.

Section 4, l 530:  Wording: The time period 2009-2019 covers the entire solar cycle 24, not just the "solar minimum".

This has now been corrected (revised text is now in the Supplement), and we thank you for pointing this out.

Section 8, remaining carbon budget. Simply for putting these numbers into perspective, it would be helpful if in the text not only the emissions between the base year 2020 and 2022 were mentioned, but also the total historical emissions up to the base year or perhaps the total from the previous decade (i.e.  from the numbers in table 1 in section 2).

This is a good idea and we have adopted your suggestion and added the historic budget until the end of 2019 following the AR6 WGI SPM Table 2.

---

## Editor Decision (ED1)

ESSD-2023-166

Suggestions and comments in attached .pdf will emerge as well during copy-editing. You will need to compile and compare lists of mandated changes vs optional suggestions. Lists might provide head-start.

References contain many strange sequences of characters. I marked only a few. These likely arise from use of multiple bioblimetric systems. Please check carefully to resolve. Particularly check that all references to IPCC AR6 chapters include valid DOI. Format slightly different supplement vs. main text? Authors will want to make their own checks of supplement references.

Check acronyms. I marked a few. Authors need to either: a) define all at first use (as often done but not yet for e.g. DAMIP); or b) include a table of acronym definitions. Authors to decide.

Small but possibly useful change: convert names and labels of figures and tables in supplement to better link to main text and to clarify associations with relevant sections. Find details in attached list.

Overall: good idea, excellent execution, thank you for using ESSD.

Page 1, line 5: Bradley Hall affiliation superscript should read as '23', not '22'. Otherwise line 38, NOAA GML, exists as an orphan.

Page 1, line 28: this institution, even if private, probably needs a country.  Or, state and country?

Page 1, line 37: Need more info about this institution?

Page 1, line 42: DKRZ needs city (Hamburg) and country (Germany). DKRZ also hosts WDCC; relevant?

Page 2, line 1: need a country?

Page 2, line 54: Affiliation #36 should have identical format to #16. One at NOAA NCEI Silver Spring, other at NOAA NCEI Asheville?

Page 2, line 58: Affiliation #40 (CIRES) not used by any author?

Page 2, line 68: "… update … to produce updated …". Need better wording here?

Page 3, lines 80-82: redundant text here

Page 3, line 82-82: "… continued series of these annual updates over this critical decade …" Good, but which decade exactly. The most recent decade? The most recent decade for which you have data? The upcoming decade? This reader knows, but many will not; need clarity and specificity. Discussed (more than adequately) in Conclusions section, but first hinted at here?

Page 3, line 103 " … global surface _**air**_ temperature?

Page 3, line 107: about FAIR: findable, accessible, reusable - yes. Interoperable: doubtful.

Page 5, line 151-154: Good paragraph! Could serve not only as introductory, as here, but also for summary! Make it more prominent?

Page 11, line 306: Authors have previously referred to generic climate models but have not yet used, nor defined, the CMIP acronym. Because you will use the acronym multiple times, it needs definition?

Page 13, line 349: Here authors use term "AR6 Working Group I" while in many other locations authors refer to WGI. Quick search and replace should ensure better consistency?

Page 14, line 363: 'AGAGE' acronym used often in this section but never defined?

Page 14, line 383: N2O uncertainty, in version I see, appears as italicized?

Page 20, line 504: Do readers need quantitative interpretation of IPCC '*very likely*' here?

Page 22, line 531-532: some confusion introduced here? IPCC 'high confidence' for EEI means but only 'very likely' for ranges? Or, high confidence for earlier time period (1976-2006) but only very likely for 2006-2018?

Pages 24, 25: Figure 4a and 4b referenced before Table 5, but Table 5 presented before Figure 4. Need a change of order, or, did I miss something? Figure 4b called out in text but Figure 4 generally or Figure 4a never? Fig 4a comes from von Schuckmann? Fig 4b might prove more informative if sorted by starting year?

Page 27, line 652-653: parenthesis missing in here somewhere?

Page 32, line 744: "also known as the carbon budget" Eliminate this phrase from this sentence, to reduce confusion with following definition?

Page 32, line 754: need altered punctuation here? E.g. replace comma with semicolon? "… systematic annual update; the decade-long period …"

Page 32, line 756: 'up-to-date'?

Page 33, Table 7, two questions: a) under 1.5C, 3rd row ('as above with AR6 update'), why do these values prove identical to those in row above? Shouldn't these values appear slightly lower than those above? b) under 2.0C, 3rd row ('as above with AR6 WGIII scenario update'), why does this text differ from equivalent rows above under 1.5 and 1.7. From line 774, doesn't reader conclude that authors applied AR6 WGIII scenario updates in all cases?

Page 34, lines 803-804: "50% RCB is expected to be exhausted a few years before the 1.5C global warming level is reached", does this occur because formal excedence of 1.5C requires decadal averaging of GMST or because of some other systematic lag? On page 40, authors remark about need to also understand non-CO2 warming. Does that reason apply here?

Page 35, line 807: depends on reader's viewpoint. Better to claim 'among' rather than 'to'?

Page 38, line 871: link works for me, thanks for including.

Page 45, line 954: Table 1 shows emissions. Reader needs to interpolate and average to estimate that rates of increases might have slowed. Perhaps to temper this conclusion

somewhat? Data would allow such an interpretation? Such a statement would better support your earlier conclusion about need to continue monitoring and metadata records?

Page 57, line 1346: redundant issue number (23) in Purkey and Johnson J. Climate article.
Supplement

Page 1, Table 1: Good to move to supplement. Clear orderly presentation. Do we need a reference, e.g. so that reader can follow to find details of e.g. HCFC-22? Perhaps original IPCC chapter?

Section 4 = okay.

Section 5 = okay.

Section 7 = okay. Suggest possible improvements: Figure numbers could perhaps change to S7.1, S7.2, etc. to better reflect inclusion in supplement and place in Section S7?

Section 8 = clarifications needed, as follows:

Page 15, line 343: "provided in Supplementary Table 1". Supplement Table 1 does not show non-CO2 reductions. Text instead refers to supplement Table 2? Another reason (see Section 7 comment above) for assigning table and figure numbers to specific sections. E.g. in this case one would refer to Figure S8.1. Earlier table would, by these numbering methods, register as Table S3.1? If, eventually, one might choose to refer to supplement tables or figures in main text, this enumeration system could provide good clarity?

Section 9 = okay but suggest change of figure label to Figure S9.1.

Page 22, line 519: This Smith et al. reference to the WGI report should have a DOI? E.g. as in the Szopa reference following soon after?

Page 23, line 544: Something strange here?